# Structures of influenza A and B replication complexes give insight into avian to human host adaptation and reveal a role of ANP32 as an electrostatic chaperone for the apo-polymerase

Benoît Arragain [1], Tim Krischuns [2,4], Martin Pelosse[1], Petra Drncova[1], Martin Blackledge [3], Nadia Naffakh [2] & Stephen Cusack [1] ✉

Replication of influenza viral RNA depends on at least two viral polymerases, a parental replicase and an encapsidase, and cellular factor ANP32. ANP32 comprises an LRR domain and a long C-terminal low complexity acidic region (LCAR). Here we present evidence suggesting that ANP32 is recruited to the replication complex as an electrostatic chaperone that stabilises the encapsidase moiety within apo-polymerase symmetric dimers that are distinct for influenza A and B polymerases. The ANP32 bound encapsidase, then forms the asymmetric replication complex with the replicase, which is embedded in a parental ribonucleoprotein particle (RNP). Cryo-EM structures reveal the architecture of the influenza A and B replication complexes and the likely trajectory of the nascent RNA product into the encapsidase. The cryo-EM map of the FluB replication complex shows extra density attributable to the ANP32 LCAR wrapping around and stabilising the apo-encapsidase conformation. These structures give new insight into the various mutations that adapt avian strain polymerases to use the distinct ANP32 in mammalian cells.

In the nucleus of the infected cell, influenza polymerase (FluPol) uses the viral genomic RNA (vRNA) as template to perform synthesis of either capped and polyadenylated viral mRNA (transcription), or unmodified anti-genomic copies (complementary or cRNA), which then serve as the template for vRNA synthesis (replication)[1,2]. The functional context for both processes is the viral ribonucleoprotein complex (RNP), a flexible supercoiled rod-shaped particle, in which the influenza virus genome is packaged by multiple copies of the viral nucleoprotein (NP) with one FluPol bound to the conserved 3' and 5' ends of either the vRNA (vRNP) or the cRNA (cRNP). Both processes require FluPol to recruit essential host factors. For transcription, FluPol binds to cellular RNA polymerase II (Pol II) to gain access to nascent, capped transcripts from which capped transcription primers are excised, a process known as cap-snatching[3,4]. In particular, FluPol binding to the serine 5 phosphorylated C-terminal domain of Pol II (Pol II pS5 CTD) is conserved amongst FluPolA, B and C although the binding sites are divergent[3,5,6]. It has been recently proposed that the Pol II pS5 CTD may serve as a platform for both transcription and replication[7], but additionally for replication, the highly conserved acidic nuclear protein 32 (ANP32) is an obligatory host factor[8,9].

[1]European Molecular Biology Laboratory, Grenoble, Cedex 9, France. [2]Institut Pasteur, Université Paris Cité, CNRS UMR3569, RNA Biology of Influenza Virus, Paris, France. [3]Institut de Biologie Structurale, Université Grenoble-Alpes-CEA-CNRS UMR5075, Grenoble, France. [4]Present address: Heidelberg University, Department of Infectious Diseases, Virology, Schaller Research Group, Heidelberg, Germany. ✉e-mail: cusack@embl.fr

ANP32 comprises a folded, N-terminal leucine-rich repeat (LRR) domain followed by a Glu-, Asp-rich intrinsically disordered region known as the low complexity acidic region (LCAR). ANP32 proteins have multiple cellular functions, notably as histone chaperones[10]. Of the three functional human ANP32 (hANP32) isoforms, hANP32A and hANP32B support human adapted influenza A (FluA) and B (FluB) virus replication[11–14], but not hANP32E[15]. ANP32 is required for both vRNA to cRNA and cRNA to vRNA replication[16,17]. It is thought that ANP32 plays at least two mechanistic roles. First, it stabilises the formation of the replication complex, an asymmetric FluPol dimer comprising a replicase, which is part of a parental RNP and synthesises the genome copy and an encapsidase, a newly synthesised apo-FluPol, which binds the 5' end of the nascent RNA replicate to nucleate formation of a progeny RNP. Second, ANP32 is proposed to recruit successive NPs to the replication complex via a direct interaction between the LCAR and NP thus facilitating co-replicational packaging of product RNA into a progeny RNP[18,19]. Extensive biochemical and mutagenesis studies have previously shown that hANP32A binds to FluPol[20,21] and NP[18,19]. Moreover, the cryogenic electron microscopy (cryo-EM) structure of the influenza C (FluC) replication complex, that is, the replicase-encapsidase dimer bound to ANP32, has been determined[22].

These data underlie the proposed model, but there are a number of aspects of the replication mechanism that remain unclear. Firstly, the structure of the replication complex has not been determined for FluA or FluB viruses, for which most of the biochemical and molecular virological data have been obtained. In the case of FluA, an avian specific 33 residue insertion in avian ANP32A (avANP32A) compared to hANP32A is critical to explain why avian adapted FluPolA polymerases cannot replicate in human cells[8]. Indeed, avian to human inter-species transmission necessitates adaptive mutations in the avian polymerase (typically PB2/E627K, D701N or Q591R) to be able to productively use the mammalian ANP32 for replication[23]. A complete molecular understanding behind these intriguing observations is still lacking. Moreover, given that the binding sites of the Pol II pS5 CTD on FluPolA, B and C are significantly different, it is likely that there has been co-evolution in the mode of ANP32 binding since the divergence of FluA, B and C. Therefore, it is of particular importance to characterise structurally the FluA and FluB replication complexes.

As a step towards further understanding of the role of ANP32 in replication, we first analysed binding of hANP32A to FluPolA and FluPolB. We show that, at least in vitro, it acts like an electrostatic chaperone[24] to stabilise apo-FluPol dimers at physiological salt concentrations (~150 mM NaCl), with distinct roles for the LRR and LCAR domains. In the case of apo-FluPolB, cryo-EM analysis reveals a previously undescribed apo-dimer structure with a 2-fold symmetrical interface, distinct from that of the previously described FluPolA dimer[25–27], with one monomer being preferentially in the encapsidase configuration. Additional cryo-EM structures (summarised in Table 1) show that hANP32A is an integral part of the FluA (strain A/Zhejiang/DTID-ZJU01/2013(H7N9)) and FluB (strain B/Memphis/13/2003) replication complexes. These structures reveal that there are significant differences in the contacts between the replicase, the encapsidase and hANP32A and in domain orientations, compared to the previously published FluC replication complex. Moreover, in the FluB replication complex, additional density clearly suggests the trajectory of the LCAR wrapping around the encapsidase. We also provide minigenome data that combined with extensive existing information in the literature validate these structures as functionally relevant. Importantly, they also provide a rationale for many of the observed mutations that favour adaption of avian FluPol to mammalian cells and that have eluded a full explanation for many years. Furthermore, these results are particularly relevant in the light of the pandemic threat posed by currently circulating, highly pathogenic avian strains, particularly A/H5N1 clades 2.3.2.1c and 2.3.4.4b. These viruses have infected diverse wild and domestic animals, including recently cows[28,29], as well as humans, with often a high mortality rate[30]. Finally, we propose a generalised trimer model of replication, whereby an ANP32-stabilised incoming apo-FluPol dimer, distinct for FluPolA and FluPolB, interacts with a replication competent vRNP or cRNP to form the functional and dynamic replication complex.

## Results

### In vitro, human ANP32A acts as an electrostatic chaperone for apo-influenza A and B polymerases

Recently it has been shown that proteins DAXX and ANP32 act as 'electrostatic' chaperones that exhibit disaggregase activity dependent on extensive polyAsp/Glu stretches within their sequences[24]. Such chaperones bind to basic peptides on the target protein and have a maximal effect between 25 and 150 mM NaCl, declining in activity between 150 and 300 mM, indicative of electrostatic interactions. To investigate whether hANP32A acts as a chaperone for apo-FluPol, we analysed by size-exclusion chromatography (SEC) and mass photometry[31] the interaction between recombinant apo-FluPolA/H7N9 or apo-FluPolB/Memphis with full-length and truncated variants of hANP32A as a function of NaCl concentration (Fig. 1; Supplementary Fig. 1).

For this study, we chose to use a functional, monomeric mutant of FluPolA for the following reasons. Apo-FluPolA forms dimers with a 2-fold symmetrical interface mediated by loops from the cores of each of the three subunits[25–27]. The peripheral PA endonuclease (PA-ENDO) and PB2 C-terminal (PB2-C) domains remain flexible in cryo-EM structures[26,27], but take up the replicase conformation when constrained in a crystal[26]. It has been proposed that, for FluPolA, template realignment following internal initiation of cRNA to vRNA replication would be specifically facilitated by transient trimer formation involving a third apo-polymerase interacting via the symmetrical dimer interface with the replicase moiety of the replication complex[22,26,32]. However, loop mutations that abolish FluPolA symmetrical dimerisation are not detrimental to FluPol activity in the minigenome assay[7]. Furthermore, they are selected for in infected cells when virus evolves to use the usually non-permissive hANP32E, in the absence of hANP32A and hANP32B[15]. It was concluded from the latter work that optimal virus replication requires the correct balance between competing symmetric and asymmetric FluPolA dimer formation, consistent with a previous study[32]. To characterise structurally the FluA asymmetric replication complex, without interference from the competing symmetric dimer, we therefore used in the following analysis the monomeric FluPolA/H7N9-4M mutant, bearing the PA/E349K, PA/R490I, PB1/K577G and PB2/G74R mutations. This mutant was demonstrated to be active in vitro and in cells in previous studies aimed at elucidating the role of the Pol II pS5 CTD in replication[7].

In Fig. 1, we present SEC experiments performed at μM FluPol concentration and complementary mass photometry measurements performed at nM FluPol concentration. Different NaCl concentrations ranging from 500 to 100 mM have been tested in the presence or absence of hANP32A, for apo-FluPolA/H7N9-4M and apo-FluPolB. In Supplementary Fig. 1, additional experiments at 150 mM NaCl compare the effect of full-length hANP32A (1-249), the LRR domain alone (1-149), the LCAR alone (144-249), or the LRR domain with half the LCAR (1-199).

In SEC, full-length hANP32A is unable to bind apo-FluPolA/H7N9-4M at 500 or 300 mM NaCl but does so at physiological salt concentrations (150–100 mM NaCl), resulting in a broadening and a shift in the elution profile (Fig. 1B, D, F, G, left, H). Complementary mass photometry, shows that, in presence of hANP32A, apo-FluPolA/H7N9-4M remains mainly monomeric at all NaCl concentrations, with a small fraction of dimers at 150 and 100 mM (Fig. 1B, D, F, G, right). In the absence of hANP32A, apo-FluPolA/H7N9-4M is soluble and monomeric at 500 and 300 mM NaCl, whereas at 150 mM, it precipitates prior to SEC although mass photometry indicates the presence of soluble apo-FluPolA/H7N9-4M monomers (Fig. 1A, C, E).

**Table 1 | Summary of FluPolA, FluPolB, and replication complex structures**

| Structure No. | Short name | Resolution | PDB/EMDB |
|---|---|---|---|
| FluPolA/H7N9-4M monomers | | | |
| 1 | Core 1 with ENDO(R) | 2.77 Å | PDB 8RMP, EMD-19366 |
| 2 | Core 2 with ENDO(R) | 2.54 Å | PDB 8RMQ, EMD-19367 |
| FluA replication complex with hANP32A | | | |
| 3 | Focus Replicase | 3.21 Å | PDB 8RMS, EMD-19369 |
| 4 | Focus Encapsidase + 627(R) + hANP32A | 3.13 Å | PDB 8RNO, EMD-19382- |
| 5 | Complete replication complex | 3.25 Å | PDB 8RMR, EMD-19368 |
| FluPolB monomers | | | |
| 6 | Apo-FluPolB encapsidase | 2.89 Å | PDB 8RN2, EMD-19384 |
| 7 | FluPolB with 5′ cRNA | 3.64 Å | PDB 8RN1, EMD-19383 |
| FluPolB symmetrical dimers | | | |
| 8 | Focus Encapsidase moiety | 2.75 Å | PDB 8RN3, EMD-19385- |
| 9 | Focus ENDO(T) moiety | 2.87 Å | PDB 8RN4, EMD-19386 |
| 10 | Focus ENDO(R) moiety | 2.88 Å | PDB 8RN5, EMD-19387 |
| 11 | Focus ENDO(E) moiety | 2.82 Å | PDB 8RN6, EMD-19388 |
| 12 | Focus core moiety | 3.09 Å | PDB 8RN7, EMD-19389 |
| 13 | Complete dimer | 2.92 Å | PDB 8RN8, EMD-19390 |
| FluPolB trimer with hANP32A | | | |
| 14 | Focus Replicase | 3.31 Å | PDB 8RN9, EMD-19391- |
| 15 | Focus Encapsidase+ hANP32A | 3.13 Å | PDB 8RNB, EMD-19393 |
| 16 | Focus Replication Complex | 3.52 Å | PDB 8RNC, EMD-19394 |
| 17 | Complete Trimer | 3.57 Å | PDB 8RNA, EMD-19392 |

When only the hANP32A LRR domain (1-149) is used at 150 mM NaCl, apo-FluPolA/H7N9-4M again precipitates prior to SEC (Supplementary Fig. 1B left, E), but mass photometry shows mainly monomers with 8% dimers (Supplementary Fig. 1B right). With the LRR with half the LCAR (1–199) at 150 mM NaCl, binding, solubilisation and a small fraction of dimers (7%) is observed as for the full-length hANP32A but the SEC profile does not shift (Supplementary Fig. 1A, C, E). With the LCAR alone (149-249) at 150 mM NaCl, binding and a shifted SEC profile similar to full-length hANP32A are observed, but no dimers are detected in mass photometry (Supplementary Fig. 1A, D, E).

Overall, these results show that only below 300 mM NaCl can hANP32A bind and solubilise apo-FluPolA/H7N9-4M at μM concentration and this depends on the presence of at least half the LCAR. Binding of the LCAR, either alone or with the LRR, to apo-FluPolA/H7N9-4M is mainly responsible for the broadened and shifted SEC profile. In addition, apo-FluPolA/H7N9-4M dimer formation requires the LRR, the LCAR alone being insufficient. These dimers are rare, and we show below that they correspond to the FluA asymmetric replication complex, consistent with the fact that the monomeric FluPolA/H7N9-4M mutant does not form symmetric dimers.

For apo-FluPolB, complementary results are obtained. In the absence of hANP32A, the protein is soluble and mainly monomeric at 500 and 300 mM NaCl (Fig. 1I, K). At 150 mM NaCl, apo-FluPolB alone precipitates prior to SEC, but mass photometry shows that 14% dimers are formed (Fig. 1M). In the presence of hANP32A, at 500 mM NaCl, apo-FluPolB is monomeric and does not co-elute with hANP32A (Fig. 1J). It only binds to hANP32A at 300 and 150 mM NaCl, forming a monomer-dimer mixture with respectively 22 and 47% dimers, as also indicated by the SEC profile shift (Fig. 1L, N, P). At 150 mM NaCl, apo-FluPolB forms significantly more dimers in the presence of hANP32A (47%) than in its absence (14%) (Fig. 1M, N), while at 100 mM NaCl, apo-FluPolB is again mainly monomeric (Fig. 1O).

Using hANP32A truncated constructs at 150 mM NaCl, the LCAR alone (144-249) stabilises apo-FluPolB dimer (46%) at a similar level to the full-length hANP32A (Supplementary Fig. 1F, I, J). With only the LRR domain (1-149), apo-FluPolB precipitates although mass photometry

detects a small fraction (7%) of dimers (Supplementary Fig. 1G, J). Finally, with the LRR plus half the LCAR (1-199), partial apo-FluPolB solubilisation is achieved with 15% dimers formed (Supplementary Fig. 1H, J).

These results shows that hANP32A LCAR is sufficient to stabilise the apo-FluPolB dimer at 150 mM NaCl concentration. A significantly smaller fraction of dimers can exist without hANP32A under these conditions at low protein concentration. Below we show by cryo-EM that this dimer is a novel symmetric FluPolB dimer, quite distinct from that of FluPolA.

For completeness, we also present results for the wild-type Flu-PolA/H7N9 (FluPolA/H7N9-WT). As previously established[26,27], and different from apo-FluPolB, apo-FluPolA/H7N9-WT without hANP32A forms equally symmetric dimers and monomers at 500 and 300 mM NaCl (Supplementary Fig. 2A, B, F). At 150 mM NaCl, the protein alone is not soluble in SEC (Supplementary Fig. 2C left, F) but mass photometry indicates a mixture of monomers, dimers and tetramers[25] (Supplementary Fig. 2C right). In the presence of hANP32A at 150 mM NaCl, apo-FluPolA/H7N9-WT is fully soluble and mainly dimeric (85%) (Supplementary Fig. 2D, F). On the other hand, vRNA promoter bound FluPolA is mainly monomeric at low NaCl even in the absence of hANP32A (Supplementary Fig. 2E, F). These results show that binding of hANP32A to apo-FluPolA/H7N9-WT at physiological salt concentrations promotes soluble symmetrical dimer formation.

Taken together, these results show that hANP32A binds to and stabilises dimers of apo-FluPolA/H7N9-WT and apo-FluPolB at physiological salt concentrations and that this depends mainly on the LCAR. Furthermore, hANP32A acts as a disaggregase for apo-FluPol at μM concentration only at physiological salt concentrations, consistent with it acting as an electrostatic chaperone, at least in vitro. We think that these ANP32 bound dimers are likely a major form of apo-FluPol in the nucleus, prior to any encounter with the RNPs that are already present, whether or not ANP32 plays an essential chaperone role in vivo.

## Human ANP32A binding to apo-influenza A and B polymerases promotes formation of the replication complex

In the preceding analysis, a minor fraction of hANP32A-bound FluPolA/H7N9-4M dimers are formed at 150 mM NaCl, this requiring at least the

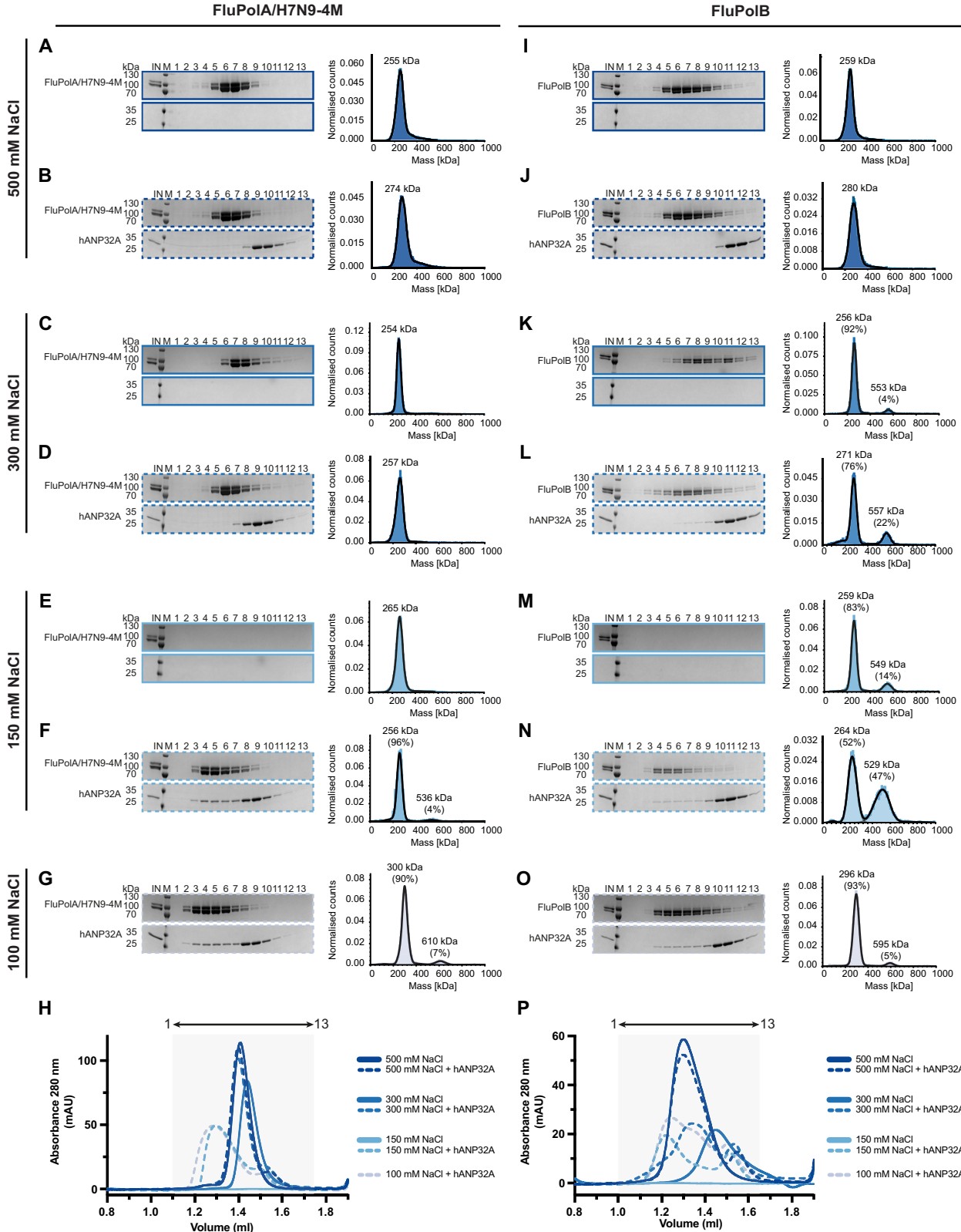

LRR domain (1-149). To characterise structurally the FluA asymmetric replication complex we analysed complexes of apo-FluPolA/H7N9-4M with hANP32A by cryo-EM (Supplementary Notes 1, 2; Table 1; Supplementary Table 1). As expected, the majority of particles are FluPolA/H7N9-4M monomers, exhibiting the PA-ENDO in the replicase conformation (ENDO(R)). Two distinct such structures were determined at 2.77 (CORE-ENDO(R)-1) and 2.54 (CORE-ENDO(R)-2) Å resolution

respectively, differing in the degree of polymerase core opening (Supplementary Notes 1; Supplementary Table 1). Consistent with the biochemical and biophysical analysis, a small fraction of particles corresponds to the FluA replication complex, comprising a replicase and an encapsidase bridged by hANP32A. The overall map resolution is limited by the flexibility between the replicase and hANP32A-encapsidase moieties as well as the presence of a preferred

**Fig. 1 | Biochemical and biophysical analysis of the interaction of FluPolA/ H7N9-4M and FluPolB with hANP32A.** SDS-PAGE and mass photometry analysis of FluPolA/H7N9-4M at 500 mM NaCl without (**A**) or with (**B**) hANP32A. The molecular ladder (M) in kDa, FluPolA/H7N9-4M heterotrimer and hANP32A are indicated on the left of the gel. 'IN' corresponds to the input. SDS-PAGE and mass photometry analysis of FluPolA/H7N9-4M at 300 mM NaCl without (**C**) or with (**D**) hANP32A. SDS-PAGE and mass photometry analysis of FluPolA/H7N9-4M at 150 mM NaCl without (**E**) or with (**F**) hANP32A. **G** SDS-PAGE and mass photometry analysis of FluPolA/H7N9-4M-hANP32A interaction at 100 mM NaCl. **H** Superposition of size exclusion chromatography (SEC) profiles of FluPolA/H7N9-4M alone (solid lines), with hANP32A (dotted lines), at 500-300-150-100 mM NaCl. Curves are respectively coloured from dark to light blue, as indicated. The relative absorbance at 280 nm (mAU) is on the y-axis. The elution volume (ml) is on the x-axis, graduated every 50 µl. SDS-PAGE fractions 1–13 corresponds to the elution volume 1.1–1.75 ml, represented as an arrow on top. SDS-PAGE and mass photometry analysis of Flu-PolB at 500 mM NaCl without (**I**) or with (**J**) hANP32A. The molecular ladder (M) in kDa, FluPolB heterotrimer and hANP32A are indicated on the left of the gel. 'IN' corresponds to the input. SDS-PAGE and mass photometry analysis of FluPolB at 300 mM NaCl without (**K**) or with (**L**) hANP32A. SDS-PAGE and mass photometry analysis of FluPolB at 300 mM NaCl without (**M**) or with (**N**) hANP32A. **O** SDS-PAGE and mass photometry analysis of FluPolB-hANP32A interaction at 100 mM NaCl. **P** Superposition of SEC profiles of FluPolB alone (solid lines), with hANP32A (dotted lines), at 500-300-150-100 mM NaCl. Curves are respectively coloured from dark to light blue, as indicated. The relative absorbance at 280 nm (mAU) is on the y-axis. The elution volume (ml) is on the x-axis, graduated every 50 µl. SDS-PAGE fractions 1–13 corresponds to the elution volume 1.0–1.65 ml, represented as an arrow on top. Source data are provided as a Source Data file (n = 1–3 independent experiments).

orientation. To alleviate this, a smaller number of particles were selected to equilibrate the distribution of orientations, giving a final map of the FluA replication complex at 3.25 Å resolution. Focussed refinement on the separate replicase and hANP32A-encapsidase moieties further improved map quality to 3.21 and 3.13 Å resolution, respectively, allowing a relatively complete model to be built (Supplementary Notes 2, 3; Supplementary Table 1), with the replicase core being similar to that in the CORE-ENDO(R)-1 structure.

Cryo-EM grids made after mixing apo-FluPolB and hANP32A show a majority of dimers with a 2-fold symmetric interface, which are quite distinct from those of FluPolA (Supplementary Notes 4, 5; Table 1; Supplementary Table 3). Several different pseudo-symmetric dimer structures were determined at 2.8 to 3.1 Å overall resolution, almost invariably comprising an encapsidase paired with a variable partner (Supplementary Notes 4). A minority of particles are apo-FluPolB trimers, whose overall structure was determined at 3.57 Å resolution. These particles comprise the FluB replication complex, which is similar to that of FluA, with an additional third polymerase forming a FluPolB symmetrical dimer interface with the replicase. Further refinement focussed on the replicase or hANP32A-encapsidase moieties improved the map quality and estimated resolution to 3.3 Å, enabling relatively complete models to be built (Supplementary Notes 6, 7; Table 1; Supplementary Table 2).

### Structure of the influenza A replication complex

The FluA replication complex comprises the asymmetric replicase-encapsidase dimer with bound hANP32A (Fig. 2; Supplementary Fig. 3). Both replicase (related domains will be annotated with (R)) and encapsidase (related domains will be annotated with (E)) have the conserved polymerase core, comprising PA C-terminal domain (PA-C), PB1 subunit and PB2 N-terminal domain (PB2-N), but with differently disposed peripheral domains (PA-ENDO and PB2-C) (Fig. 2A; Supplementary Fig. 3B, C). Compared to the transcriptase conformation (Supplementary Fig. 3A), the replicase conformation is characterised by a rotated PA-ENDO(R), against which packs the PB2 nuclear localisation signal domain (PB2-NLS(R)), with the C-terminal, helical NLS containing peptide extending across the ENDO surface[26,33,34](Fig. 2A; Supplementary Fig. 3A, B, D). The PB2 cap-binding domain (PB2-CBD(R)) is packed against the palm domain of PB1(R) (Supplementary Fig. 3E). The PB2/627-NLS(R) double domain is in the open conformation[35] with the linker extended and the otherwise flexibly connected PB2-627(R) domain being held in place in the FluA replication complex by interactions with the PB2-NLS(E) domain (Fig. 2A). In the distinct encapsidase conformation (Supplementary Fig. 3C), the PA-ENDO(E) packs on the PB1(E) fingers domain, but has only low-resolution density. The flexible PA-ENDO(E) 51-72 insertion contacts the top of the PB2-CBD(E) (e.g. residues PA/55-57 with PB2/I461, 469-471, K482), which is not rigidly integrated into the FluA replication complex either (Fig. 2A; Supplementary Fig. 3C, F). Interestingly, this PA loop, found in FluPolA and FluPolB but not FluPolC, has been shown

to be essential for replication in the case of FluA[36]. The PB2(E) midlink domain is stabilised in position by residues 520-524 forming an anti-parallel alignment of strands with PB2-N(E) residues 126-132 (Supplementary Fig. 3C, G). PB2-N(E) residues 138-226, which include the helical lid domain, are not visible in the map, having apparently been displaced to avoid clashing with the PA-ENDO(E). There is putative density for the PB1-C(E)/PB2-N(E) helical interface bundle, but no model can be built, contrary to the situation in the FluB replication complex (see below). The PB2/627-NLS(E) double domain is in the closed conformation with the 627-domain packing against PA-C(E). The PB2-NLS(E) domain makes a substantial contact with the PB2-627(R) domain (Fig. 2A).

The interface between the replicase and the encapsidase buries a solvent accessible surface of around 3300 Å², with three main zones of contact (Supplementary Fig. 4). The first involves PB2-N(R) β-strands (128-134, 243-250), neighbouring PA(R) 432-438 loop interacting with the PA(E) arch (N-terminal side, 368-377) and the tip of the PB1(E) β-hairpin (361-364) (Supplementary Fig. 4A, B). The latter region is close to the encapsidase 5' hook binding site, which however is empty in this apo-structure. Key hydrophobic contacts are made by PA-C(E)/ I330, W368 (which changes rotamer) and M374 to PB2-N(R)/T129, M243 and T245; PB1(E)/M362 to PA(R)/P434 and I438, and PB1(E)/K363 to PB2-N(R)/F130 (Supplementary Fig. 4B). The impact of various mutations designed to disrupt this interface was tested using cell-based assays for the FluPolA/WSN/33 in a vRNP reconstitution assay with a vRNA reporter to assess overall transcription/replication activity and a split luciferase-based complementation assay to assess binding to hANP32A (Supplementary Fig. 4C-E). FluPol activity was significantly impaired in the presence of the PA/I330A and PB2/T129A, T245A mutations, more markedly so when they were combined (Supplementary Fig. 4C), consistent with the described interactions. Similar trends were observed when FluPol activity support by either hANP32A, hANP32B or chANP32A was determined by transient complementation in HEK-293T cells knocked out for hANP32A and hANP32B (Supplementary Fig. 4D). This is consistent with decreased FluPol-binding levels to either hANP32A, hANP32B, or chANP32A, as determined in a split luciferase-based complementation assay (Supplementary Fig. 4E).

The second zone of interaction between replicase and encapsidase involves the C-terminal β-sheet region of PB2-627(R) (residues 645, 651-657, 668-669) with PA(E) (315-316, 550-loop 547-558) (Supplementary Fig. 4A, F). Notable hydrophobic interactions include PB2-627(R)/M645 and L668-G669 with PA(E)/F315, and PB2-627(R)/P654 with PA(E)/L549, together with a hydrogen bond between PA(E)/Q556 and PB2-627(R)/N652 carbonyl oxygen (Supplementary Fig. 4F). The third zone of interaction is localised at the interface of PB2-627(R) (residues 585-587, 631-637 on the 627-loop, 644-646) with PB2-NLS(E) (residues 703-708, 712, 715-720) (Supplementary Fig. 4A, G). Notable hydrophobic contacts are made by PB2-627(R)/A587, M631, F633, T637 and R646 (Supplementary Fig. 4G). Using the cell-based assays described above, we found that the mutations PB2/A587K or A717E

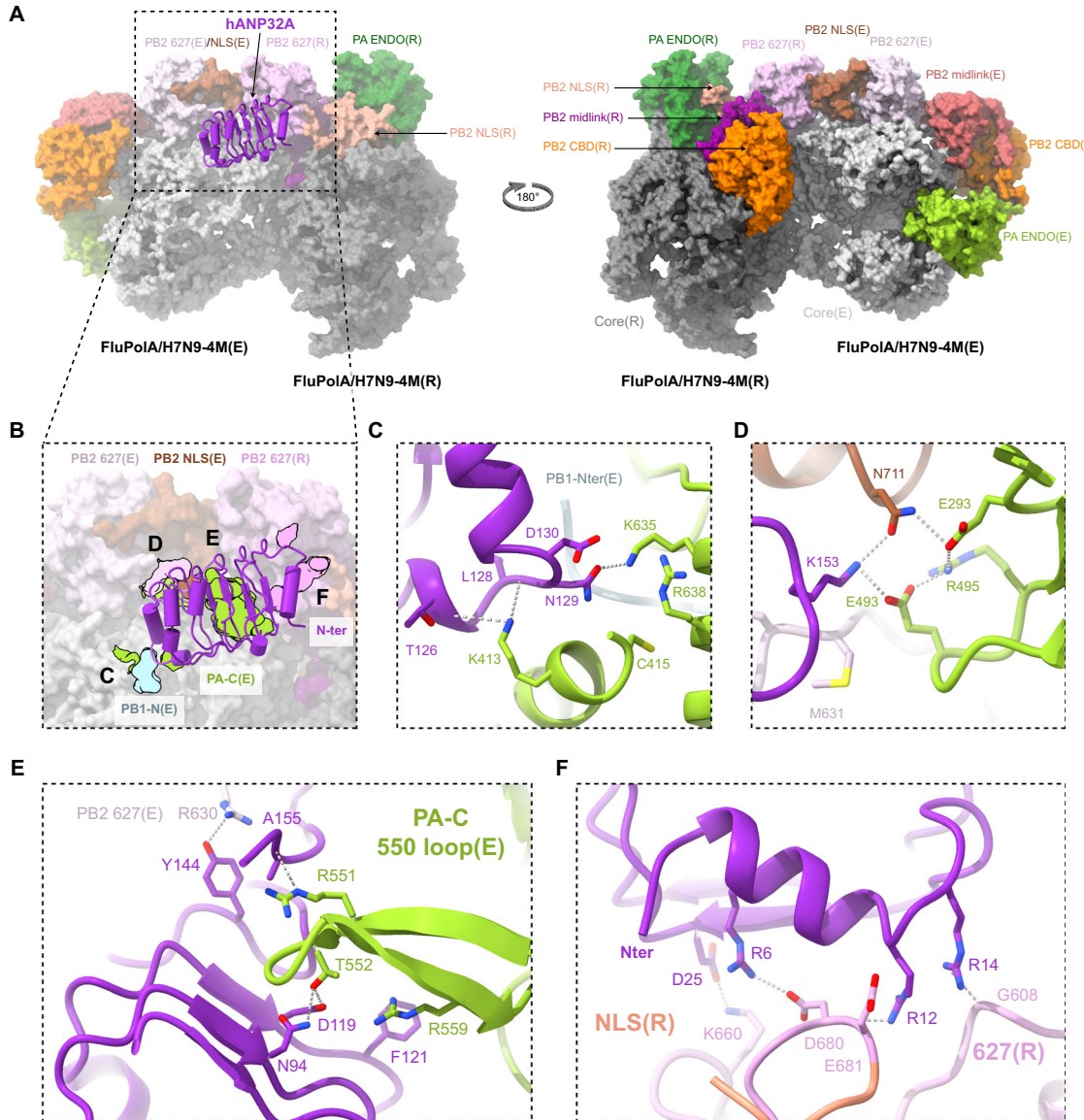

**Fig. 2 | Overall structure of the FluPolA/H7N9-4M replication complex and the interactions with hANP32A. A** Surface representation of the FluPolA/H7N9-4M replication complex with hANP32A displayed as cartoon (purple). FluPolA/H7N9-4M replicase (R) core is dark grey. PA ENDO(R) is in dark green, PB2 midlink(R) magenta, PB2 CBD(R) orange, PB2 627(R) pink and PB2 NLS(R) beige. FluPolA/H7N9-4M encapsidase (E) core is light grey. PA ENDO(E) is in light green, PB2 midlink(E) salmon, PB2 CBD(E) orange, PB2 627(E) light pink and PB2 NLS(E) brown. **B** Close-up view of hANP32A interactions with FluPolA/H7N9-4M replication complex. Interaction surface are highlighted, main contacts are labelled from (**C**) to (**F**) and coloured according to FluPolA/H7N9-4M interacting domains. PB1-N(E) is coloured in light blue, PA-C(E) in light green, with PB2 627(E)/NLS(E), PB2 627(R) coloured as in (**A**). **C** Cartoon representation of the interaction between hANP32A

128-130 loop and FluPolA/H7N9-4M PA-C(E) and PB1-N(E). hANP32A and FluPolA/H7N9-4M domains are coloured as in (**B**). Ionic and hydrogen bonds are shown as grey dotted lines. **D** Cartoon representation of the interaction between hANP32A K153 and FluPolA/H7N9-4M PA-C(E) and PB2 627(E)/NLS(E). hANP32A and FluPolA/H7N9-4M domains are coloured as in (**B**). Ionic and hydrogen bonds are shown as grey dotted lines. **E** Cartoon representation of the interaction between hANP32A curved β-sheet and FluPolA/H7N9-4M PA-C(E) 550-loop. hANP32A and FluPolA/H7N9-4M domains are coloured as in (**B**). Ionic and hydrogen bonds are shown as grey dotted lines. **F** Cartoon representation of the interaction between hANP32A N-terminus and FluPolA/H7N9-4M PB2 627(R)/NLS(R). hANP32A and FluPolA/H7N9-4M domains are coloured as in (**B**). Ionic and hydrogen bonds are shown as grey dotted lines.

significantly reduced overall FluPol activity (Supplementary Fig. 4H), its dependence on either hANP32A, hANP32B or chANP32A (Supplementary Fig. 4I) as well as binding to hANP32A, hANP32B or chANP32A (Supplementary Fig. 4J), consistent with the structural findings. The PB2/A717E mutation had the strongest effect on FluPol activity, but not on binding to hANP32A, suggesting that it impairs another FluPol function beyond the replicase-encapsidase interaction.

The PB2-627(R) domain behaves as a rigid-body part of the encapsidase in focussed cryo-EM refinement (Supplementary Notes 2), which is explained by its interfaces with PB2-NLS(E) and PA-C(E) domains. The particular juxtaposition of the PB2-627(R) domain with

the PB2/627-NLS(E) double domain is a key feature that distinguishes both the FluA and FluB replication complexes from the previously described FluC replication complex (Fig. 3). In FluPolA, the PB2-NLS(E) domain is sandwiched between the PB2-627(E) and PB2-627(R) domains, with no contact between the latter two domains, whereas in FluPolC the PB2/627-NLS(E) double domain is rotated relative to the PB2-627(R) domain by ~78° and the PB2-NLS(E) domain squeezed to one side (Fig. 3A, B). Consequently, for FluC, the replicase and encapsidase PB2/627-domain loops containing K649 (equivalent to E627 in FluPolA/H7N9 and K627 in FluPolB) are closer and face each other, with the two K649 Cα atoms being ~19 Å apart. The LCAR is

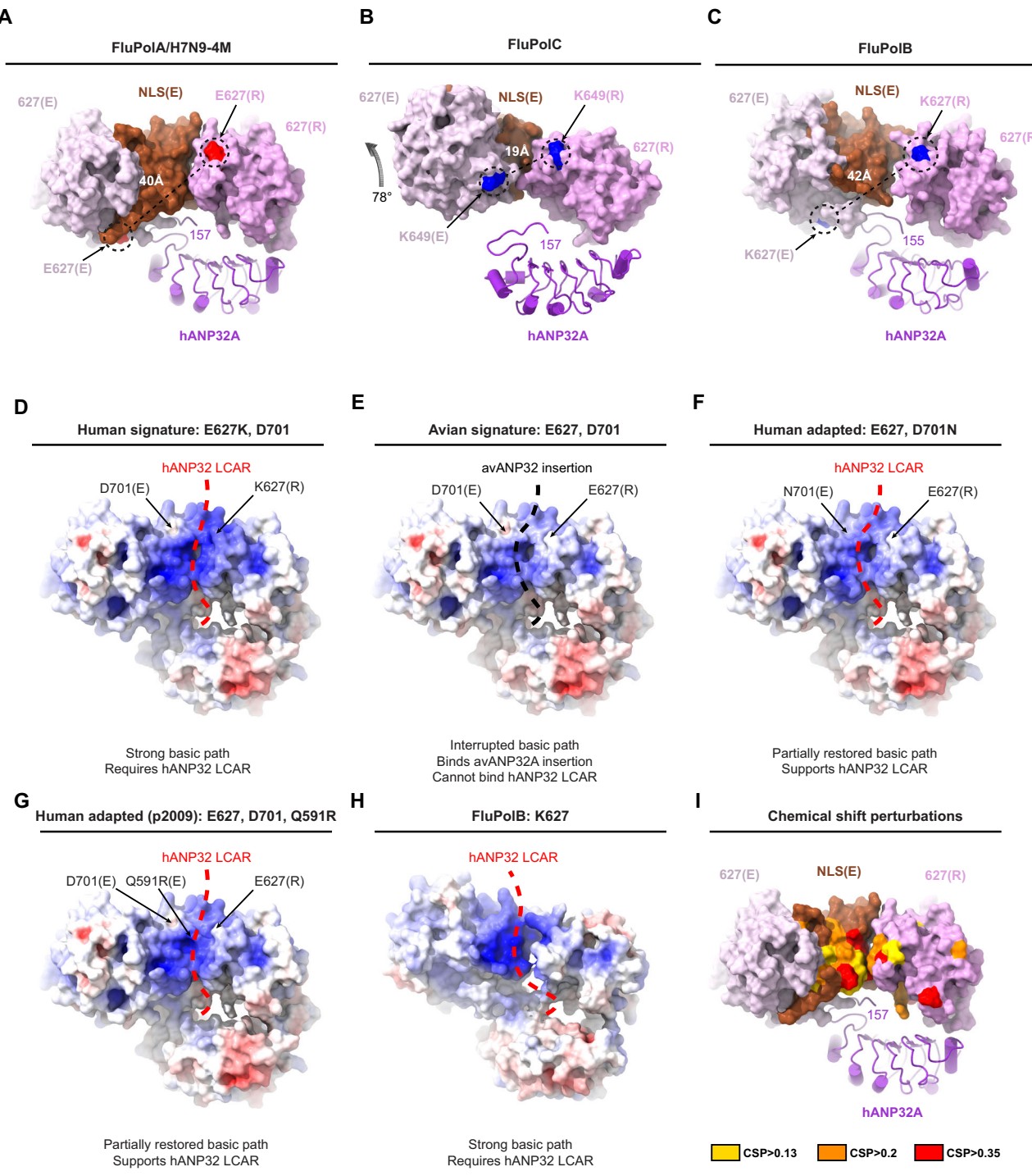

**A** FluPolA/H7N9-4M **B** FluPolC **C** FluPolB

**D** Human signature: E627K, D701

Strong basic path
Requires hANP32 LCAR

**E** Avian signature: E627, D701

Interrupted basic path
Binds avANP32A insertion
Cannot bind hANP32 LCAR

**F** Human adapted: E627, D701N

Partially restored basic path
Supports hANP32 LCAR

**G** Human adapted (p2009): E627, D701, Q591R

Partially restored basic path
Supports hANP32 LCAR

**H** FluPolB: K627

Strong basic path
Requires hANP32 LCAR

**I** Chemical shift perturbations

CSP>0.13 CSP>0.2 CSP>0.35

proposed to pass over this interface[22]. In the FluA and FluB replication complexes, these two loops are rotated far apart from each other with a main chain distance of respectively ~40 Å and ~42 Å between equivalent PB2/627 residues (Fig. 3A, C). This makes a significant difference to the surface with which the LCAR of hANP32A is likely to interact in the FluA and FluB replication complexes compared with FluC.

## Interactions of human ANP32A with the influenza A polymerase encapsidase and replicase

Binding of the hANP32A LRR domain to the FluA replication complex buries ~3300 Å² of solvent accessible surface of which 80 % is with the encapsidase (Fig. 2B). The C-terminal end of the LRR domain, notably

the 128-129 loop, packs against the PA-C(E) domain, burying the N-terminus of PB1(E) and the peptide 152-157 curves back against the LRR domain to contact the PB2/627-NLS(E) double domain (Fig. 2B). In particular, hANP32A/N129 makes a key interaction with PA(E)/K635, a residue previously shown to be critical for the binding of Pol II pS5 CTD in site 1 of both FluPolA and FluPolB[3,7]. In addition, PA(E)/K413 makes multiple hydrogen bonds to the main chain of residues 126-128 of hANP32A (Fig. 2C). Consistently, when cell-based mutational analysis was performed in the FluPolA/WSN/33 minigenome assay, the PA/K413A, PA/K413E and PA/K635A mutations reduced FluPol activity (Supplementary Fig. 5A), FluPol dependence on hANP32A, hANP32B or chANP32A (Supplementary Fig. 5B) and binding to hANP32A, hANP32B or chANP32A (Supplementary Fig. 5C). The effect is most dramatic for

**Fig. 3 | Human ANP32A, PB2 627-NLS(E) and PB2 627(R) domain organisation and implication in human adaptive mutations. A** Surface representation of FluPolA/H7N9-4M PB2 627-NLS(E) and PB2 627(R) domains. hANP32A is displayed as cartoon. PB2/E627 residue is coloured in red as surface. The distance between PB2/E627(R) and E627(E) is indicated. The last visible hANP32A C-terminal residue is annotated. hANP32A and FluPolA/H7N9-4M domains are coloured as in Fig. 2. **B** As (**A**) but for FluPolC PB2 627-NLS(E), PB2 627(R) and hANP32 as extracted from PDB 6XZQ. The FluPolC PB2 627(R) domain is aligned with the equivalent domain in (**A**). FluPolC PB2 627-NLS(E) then differs by a 78 degree rotation compared to FluPolA/H7N9-4M PB2 627-NLS(E), as indicated by the arrow. PB2 K649 residue is coloured in blue as surface. The distance between PB2/K649(R) and K649(E) is indicated. **C** As (**A**) but for FluPolB PB2 627-NLS(E) and PB2 627(R) domains after alignment of the 627(R) domain with that in (**A**). PB2 K627 residue is coloured in blue as surface. The distance between PB2 K627(R) and K627(E) is indicated. **D–G** Surface representation of FluPolA/H7N9 627-NLS(E) and 627(R) domains, bearing human or avian adapted mutations, coloured by electrostatic potential (red

negative, blue positive): (**D**) human signature with PB2 E627K/D701 (modelled); (**E**) avian signature with PB2 E627/D701 (modelled); (**F**) human adapted avian signature with PB2 E627/D701N (this study); (**G**) human adapted A/H1N1pdm09 signature with PB2 E627/D701/Q591R (modelled). In (**D, F, G**), a strong basic path favours binding of hANP32 LCAR (red dotted line). In (**E**), an interrupted basic path allows the avANP32A insertion to bind (black dotted line). **H** Surface representation of FluPolB PB2 627(E)/NLS(E) and PB2 627(R) domains coloured by electrostatic potential. The putative path of the hANP32 LCAR is shown as a red dotted line. **I** Chemical shift perturbations (CSPs), calculated from differences between free and hANP32A bound forms of 627(K)-NLS from A/duck/Shantou/4610/2003(H5N1) (from Fig. 2 of ref. 21). CSPs were mapped onto the structure of the 627-NLS(E) and 627(R) domains from the FluPolA/H7N9-4M replication complex. CSPs from the 627 domain were designated as being associated with the replicase, while shifts from the NLS domain were associated with the encapsidase. Red corresponds to the highest CSPs (CSP > 0.35), orange to intermediate CSPs (>0.2) and yellow to lower but still measurable CSPs (CSP > 0.13).

PA/K413E which shows FluPol activity and binding levels close to background. These data are in agreement with a previous report that the PA/K413A mutation affects replication of FluPolA, based on the observed role of the equivalent residue K391 in the FluC replication complex[22] (note that this residue is not conserved in FluPolB). Similarly, it has been previously reported that the PA/K635A mutant is not only defective for Pol II pS5 CTD binding and hence transcription activity, but also replication activity[3,7]. The observation that PA/K635 is important for binding to both the Pol II pS5 CTD and to hANP32A suggests that their simultaneous interaction with PA-C is sterically impossible, notably in the context of the encapsidase. Finally, hANP32A/K153 also makes interactions with PB2(E)/N711 and PA(E)/E493, which are stabilised by PA(E)/R495 and E293 in a network of polar interactions (Fig. 2D).

To assess the impact of hANP32A mutations on FluPolA activity, vRNP reconstitution were performed in HEK-293T cells knocked out for ANP32A and ANP32B, and transiently complemented with a WT or mutant hANP32A. Compared to WT, hANP32A mutants K153A and K153E, as well as the PA/E493K mutant, were less efficient in supporting FluPol activity (Supplementary Fig. 5D). Interestingly, FluPol activity was partially rescued when charge-reversal mutants PA/E493K and hANP32A/K153E (but not K153A) were combined (Supplementary Fig. 5D), indicating that the interaction is restored to some extent. Consistently, hANP32A/K153E and PA/E493K individually decreased FluPol-binding to hANP32A, but showed increased binding-levels when tested in combination (Supplementary Fig. 5E). This is in line with the observed interactions shown in Fig. 2C, D and with previous results showing that this region of hANP32A is critical for its interactions with FluPol[37].

Another major point of contact is of the PA-C(E) 550-loop, which bends to be able to interact with the concave β-sheet surface of hANP32A. Loop residues R551, T552 (an avian specific residue[38], normally serine in mammalian-adapted FluPol) and R559 make direct interactions with hANP32A/A155, D119, N94 and F121 respectively (Fig. 2E). A deletion in the PA-C 550-loop was previously shown to affect replication in a cell-based assay[6,7]. Consistent with these observations, we show that the triple mutation R551A-S552A-R559A in A/WSN/33 PA affects both FluPol activity and binding to hANP32A, as does the double mutation F121A-N122A in hANP32A, although with a relatively modest effect on FluPol activity (Supplementary Fig. 5F, G). Beyond residue 157 there is only disjointed, low resolution density for hANP32A, so that the conformation and interactions of the LCAR cannot be visualised precisely in the FluA replication complex.

The interactions of the replicase with hANP32A are more tenuous (Fig. 2F). K660 from the PB2-627(R) 660-loop makes a salt bridge with hANP32A/D25. Residues 680-DE from the extended PB2/NLS-627(R) linker could make electrostatic interactions with R6 and R12 of hANP32A and G608 from PB2-627(R) with R14, although the density is

relatively poor in this region, due to mobility. Again, cell-based mutational analysis (PB2/K660A, hANP32A/D25A) confirmed the structural findings (Supplementary Fig. 5H–I). Importantly, steady-state levels of the WT and mutant PA, PB2 and hANP32A proteins used for functional studies in cell-based assays were similar as determined by western blot (Supplementary Fig. 5J–L).

## Correspondence with published studies on influenza polymerase-ANP32 interactions

There is already abundant literature on putative interactions between hANP32A and FluPolA. Residues 129-130 have been shown to be critical for defining functional (or non-functional) species and isoform specific interactions of ANP32. In mammals, these residues are generally 129-ND in ANP32A and ANP32B, or NA or SD in mouse respectively, all of which support FluA replication, although mouse proteins are suboptimal[12,39]. Avian ANP32A has 129-ND and supports FluA replication, whereas avANP32B (129-IN) and avANP32A with the single N129I mutation do not[11]. Human or avian ANP32E, with 129-ED, poorly support replication[11,15,40]. These observations are fully consistent with the FluA replication complex structure that shows hANP32A/N129 interacting with PA(E)/K635 (Fig. 2C) and which can be plausibly substituted by the smaller serine, as in mouse, but not by the larger isoleucine or glutamate. A virus that has evolved to use hANP32E in human cells knocked out for hANP32A and hANP32B acquires the PB1/K577E and PA-C/Q556R (550-loop) mutations[15]. The PB1 mutation likely acts by weakening the competing FluPolA symmetric dimer interface as proposed[15], similar to the monomeric FluPolA/H7N9-4M mutant, which bears the PB1/K577G substitution. In addition, our FluA replication complex structure shows that the PA/Q556R mutation in the encapsidase could make a salt-bridge with hANP32A/E154 (which is conserved in ANP32E), thus promoting replication complex formation (Supplementary Fig. 6A, B). In a related experiment, when virus is selected to replicate in transgenic chickens or chicken embryos carrying mutant chicken (ch) ANP32A (N129I, D130N) instead of WT chANP32A, escape mutants PA/E349K, Q556R, T639I, G634E, K635E, K635Q and PB2/M631L, I570L are found, with a predominance of PA/E349K and PB2/M631L. PA/E349K again acts by weakening the FluPolA symmetric dimer interface[15,32], also shown by the FluPolA/H7N9-4M mutant, which bears this substitution, whilst PA/Q556R would strengthen the interaction with hANP32A (see above). The other PA mutations cluster around the key contact with the hANP32A 129-130 loop, plausibly making local perturbations that better accommodate 129-IN (Supplementary Fig. 6A). PB2/M631L, in the encapsidase, is at the other main contact where the polymerase interacts with hANP32A/K153 and again may facilitate accommodation to the mutated chANP32A/D130N (Supplementary Fig. 6A). Interestingly, PB2/M631L is the most consistently observed FluPol adaptive mutation in the recent outbreak in the US of high pathogenic avian influenza in cows,

with a very low detection of PB2/E627K and D701N[28,29]. In addition, the recently described PA/E613K mutation present in clade A in cattle-derived sequences can also be explained as strengthening the replicase-encapsidase interface based on the FluA replication complex structure (Supplementary Fig. 6E).

## The influenza A replication complex structure explains avian to human host adaptations

Adaptation of avian strain FluPols (which invariably have PB2/E627) to be able to function in human cells generally requires PB2 mutations. Real-life evolution of circulating FluA viruses and numerous laboratory studies show that the most effective routes to adapt avian FluPol to mammalian ANP32A or ANP32B are PB2/E627K[41], PB2/D701N[42,43] or PB2/Q591R (A/H1N1pdm09 strain)[23,44], with other observed mutations (PA/T572S, PB2/T271A, PB2/K702R, PB2/D740N) potentially assisting to a lesser extent.

To understand why these residues make such a difference, we calculated electrostatic surfaces using the FluA replication complex structure with appropriate modelled substitutions in replicase and encapsidase for the four cases (Fig. 3D–G): (1) typical human signature with PB2/Q591, K627, D701 (Fig. 3D), (2) typical avian signature with PB2/Q591, E627, D701 (Fig. 3E), (3) human adapted FluPolA/H7N9 with PB2/Q591, E627, N701 (corresponding to our structure) (Fig. 3F), and (4) human adapted FluPolA/H1N1pdm09 with PB2/R591, E627, D701 (Fig. 3G).

The typical human signature results in an uninterrupted positively-charged path following the PB2-NLS(E)/PB2-627(R) interface (and continuing round the back), encompassing K627(R) and skirting the acidic patch due to D701(E) (Fig. 3D). The structure of the fully human adapted FluB replication complex (see below) shows a similar strong basic path (Fig. 3H). Residues in the FluA replication complex contributing to this basic surface are PB2-NLS(E)/K702, R703, K718, K721, K738, R739 and PB2-627(R)/K586, R589, K627, R630. We propose that this is likely the trajectory followed by the proximal part of the acidic LCAR of hANP32A or hANP32B (e.g. residues 160-AEGYVEGLDDEEEDED*EEEYDEDAQVV*-186, italic here is mammalian specific), interacting in a predominantly electrostatic, multivalent fashion, since it is not clearly observed in the structure. Indeed, projecting the residues undergoing chemical shifts when hANP32A interacts with isolated 627-NLS domain, onto PB2-NLS(E)/PB2-627(R) rather than the closed PB2/627-NLS(E) conformation, highlights perturbations (due to direct or indirect binding effects) at the PB2-NLS(E)/PB2-627(R) interface[21] (Fig. 3I). The typical avian signature results in an interrupted basic track, due to the combined effect of positively charged D701(E) and E627(R) (Fig. 3E). This surface is more appropriate to bind the avANP32A due to the 33 amino acid insertion (i.e. residues 160-AEGYVEGLDDEEEDED**VLSLVKDRDDK**-186, bold here is avian specific). This could place the avian specific hexapeptide 176-VLSLVK that strongly interacts, according to NMR[21], and two other basic residues in the equivalent region.

Both human adapted signatures, as in FluPolA/H7N9 and FluPolA/H1N1pdm09, partially restore the basic track (Fig. 3F, G). Interestingly PB2/Q591R might have a dual mode of action in both replicase and encapsidase, since the simultaneous Q591R(E) mutation could enhance binding of hANP32 to the encapsidase through formation of a salt bridge with hANP32A/D151 (Supplementary Fig. 6C). Furthermore, the 'third-wave' mutation PA/N321K in FluPolA/H1N1pdm09 is thought to be an additional adaptation of a swine FluPol to hANP32[39,45]. Indeed, the mutation PA(E)/N321K could lead to a salt bridge with PB2(R)/E249 (Supplementary Fig. 6D), thus strengthening the encapsidase-replicase interface, a mechanism suggested to compensate for a sub-optimal ANP32 interaction[15].

It has also been shown that mammalian ANP32A and ANP32B proteins preferentially drive different adaptive mutations in avian FluPol, respectively PB2/D701N or PB2/E627K, and this ability maps to

the significantly different LCAR[23] (Supplementary Notes 8). One possible explanation is that hANP32B is considerably more acidic than hANP32A in the region 176-190, with an insertion of five extra acidic residues and substitution of three non-charged residues by acidic residues (Supplementary Notes 8). This hyper-acidic stretch of hANP32B may require the more basic LCAR track, resulting from the PB2/E627K mutation, to bind in a functional way.

Importantly, the FluA replication complex structure reveals a significant asymmetry in the positioning of the encapsidase PB2/627 and 701 residues and their counterparts in the replicase. Only the 627(R) and 701(E) residues are in the putative pathway of the LCAR (Fig. 3). This would suggest that the nature of these residues, whether E/K or D/N, should only exert their influence in human cells via the replicase or encapsidase, respectively. Whereas the relevant position of PB2/701 has not been analysed, several studies have addressed the effect of making the PB2/E627K substitution only in the encapsidase or only in the replicase[16,17,46]. These studies are based, firstly, on a cRNA stabilisation assay involving infection of HEK-293T cells in the presence of pre-expressed NP and PB2/627E or PB2/627K FluPol with an inactive PB1, with added actinomycin D or cycloheximide to prevent transcription/translation by the incoming vRNPs. The incoming virus thus provides the vRNA to cRNA replicase, whilst the pre-expressed FluPol acts as encapsidase. Results indicate that both incoming avian 627E and 627K viruses produce stable cRNPs in infected cells, whether the pre-expressed PB2 is 627E or 627K[16,17,46]. In a second assay, performed in the presence of pre-expressed NP and FluPol with an active PB1 (i.e. a replication assay), Manz et al.[46] found that only 627K viruses could produce functional cRNPs in human cells. In contrast, Nilsson-Payant et al.[16] and Swann et al.[17] found that: (i) cRNPs produced by 627E and 627K viruses can both serve as a template for cRNA to vRNA synthesis, provided that the pre-expressed PB2 (which now is part of the replicase) is 627K; and (ii) the impaired vRNA synthesis when the pre-expressed PB2 is 627E can be restored by pre-expressing chANP32A.

Taken together, these observations indicate that in human cells, the PB2/E627 FluPol is functional as a replicase to perform vRNA to cRNA synthesis and as an encapsidase, but is impaired as a replicase to co-opt hANP32 to perform cRNA to vRNA synthesis. This agrees with the structure showing that only the replicase 627 residue is part of the likely LCAR trajectory, however, why this restriction only affects cRNA to vRNA replication remains unexplained.

## Structure of the apo-influenza B polymerase symmetrical dimer

The biochemical and biophysical analysis revealed that a stable apo-FluPolB dimer is formed in the presence of hANP32A at physiological salt concentrations (Fig. 1). Consistently, the majority of particles in the apo-FluPolB-hANP32A cryo-EM analysis are dimers and monomers, with a minority of trimers (Supplementary Notes 4–6). The dimers have a 2-fold symmetrical interface, although the peripheral domains of each monomer can be in quite different conformations (Fig. 4; Table 1; Supplementary Table 3; Supplementary Notes 4, 5). The apo-FluPolB dimer interface involves the PA-arch residues 375-385 of one monomer contacting PA/332-338 and 361-364 of the second monomer, and vice versa. In addition, the tips of the PB1 β-hairpin of each monomer (residues 360-363, closely associated with the PA-arch), interact with each other across the 2-fold axis (Fig. 4A–C). The core dimer interface buries only ~2200 Å² of solvent accessible surface compared to ~3600 Å² for the FluPolA symmetrical dimer. The apo-FluPolB dimer interface involves hydrophobic (e.g. PA/F335, Y361, W364, I375, M376, V379), polar and salt-bridge interactions (e.g. PA/D382 with Y361 and K338, E378 with K338 and K358) (Fig. 4C). The PA peptides 357-372 and 504-513, containing aromatics residues PA/Y361, W364 and PA/H506 are refolded in the apo-form compared to their configuration in the 5' hook-bound form of FluPolB, where PA/H506 stacks on nucleotide 11 of the 5' hook (Fig. 4D). This suggests

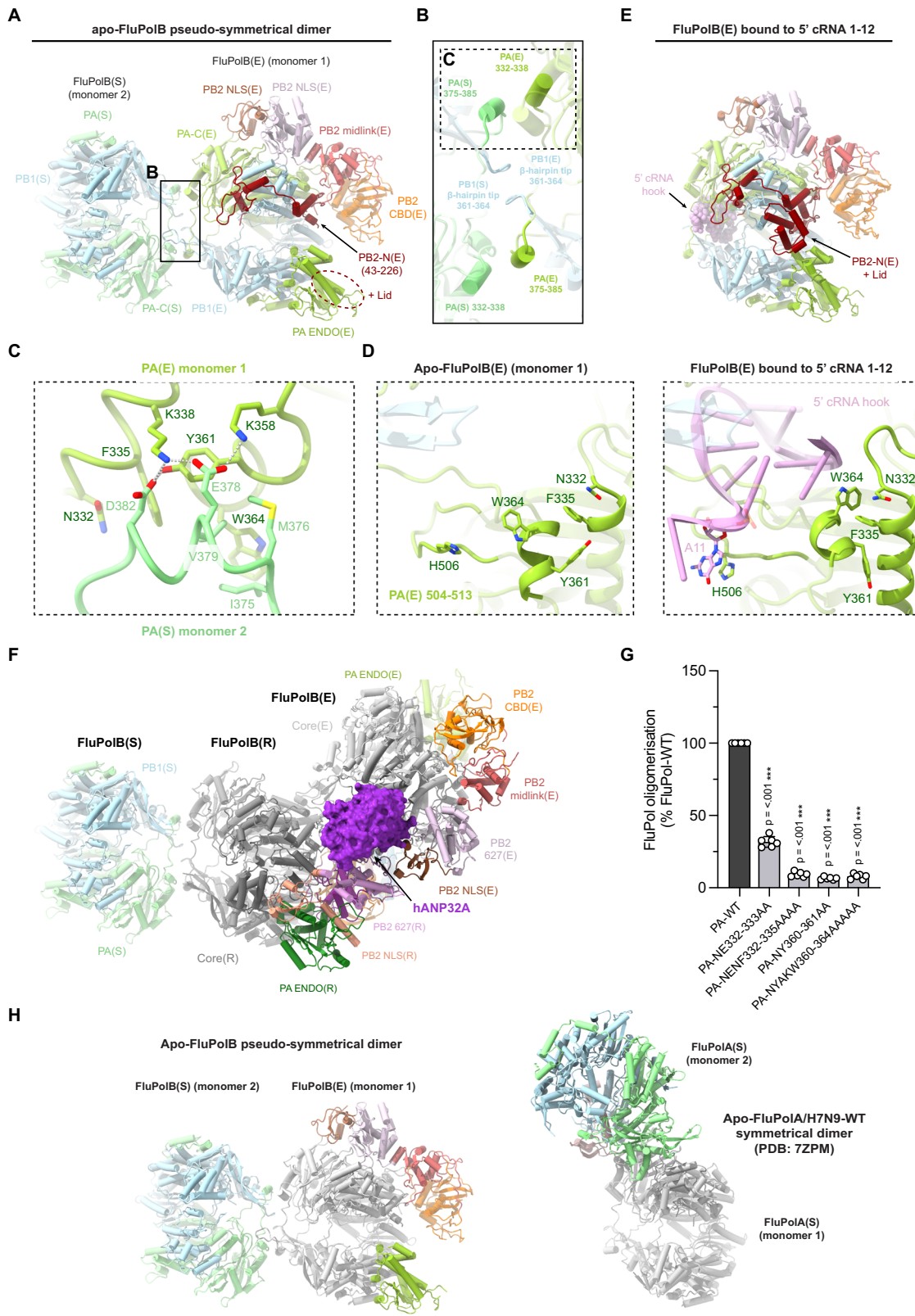

that for FluPolB, symmetrical dimer formation and 5' hook binding might be mutually exclusive. Biochemically and biophysically, this is found to be the case, since FluPolB bound to the vRNA 5' hook is soluble and monomeric at 150 mM NaCl and, furthermore, does not bind hANP32A (Supplementary Fig. 7). This highlights the fact that hANP32A is mainly required to chaperone apo-FluPol, even though we see no density corresponding to it in the apo-FluPolB

symmetrical dimer structures. We confirmed these observations by structure determination of a monomeric form of the FluPolB encapsidase bound to nucleotides 1-12 of the cRNA 5' hook (Fig. 4E; Table 1; Supplementary Table 4; Supplementary Notes 9), which can serve as a model for the vRNA replication product-bound encapsidase. We conclude that 5' hook binding disassociates the apo-FluPolB symmetric dimer or, conversely, under certain circumstances, formation

**Fig. 4 | Apo-FluPolB pseudo-symmetrical dimer and 5′ cRNA bound FluPolB encapsidase. A** Cartoon representation of the most abundant apo-FluPolB pseudo-symmetrical dimer. Monomer 1 has the encapsidase conformation, FluPolB(E) with PA-C(E) in light green, PB1(E) in light blue, PB2-N(E) (43-226) in dark red, PB2 midlink(E) in salmon, PB2 CBD(E) in orange, PB2 627(E) in light pink, PB2 NLS(E) in brown. Putative PB2-N(E) lid density (dotted ellipse) is located next to PA-ENDO(E). Monomer 2, FluPol(S), takes multiple conformation. Here, only PA(S) and PB1(S) subunits are shown, respectively in light green and light blue. The cores from each monomer form a symmetrical interface highlighted by a dotted rectangle, corresponding to (**B**). **B** Close-up view of the FluPolB symmetrical dimer interface. The main interaction is mediated by PA(E)/332-338 interacting with PA(S)/375-385. PB1 β-hairpin tips from both monomers interact with each other across the 2-fold axis. One of the symmetrical dimer interfaces is highlighted with a dotted rectangle, corresponding to (**C**). **C** Detail of the residue contacts between PA(E)/332-338 and PA(S)/375-385 at the symmetrical dimer interface. Domains are coloured as in (**A**, **B**). Ionic and hydrogen bonds are shown as grey dotted lines. **D** Structural rearrangement of PA(E) upon 5′ cRNA hook binding. The 5′ cRNA hook (nts. 1–12) is coloured plum with nucleotides as stubs. **E** Cartoon representation of the 5′ cRNA bound FluPolB encapsidase structure. Domains are coloured as in (**A**). The 5′ cRNA

hook (nts. 1–12) is displayed as spheres and coloured plum. The PB2-N(E) lid is observed when FluPol(E) is bound to the 5′ cRNA hook. **F** Cartoon representation of the complete FluPolB trimer, composed of the replication complex (FluPolB(R) +hANP32A+FluPolB(E)), with FluPolB(R) core forming a pseudo-symmetrical dimer with a third FluPolB(S). FluPolB(S) is orientated and coloured as in (**A**). hANP32A is displayed as a purple surface. FluPolB(R) core is dark grey, PA ENDO(R) is dark green, PB2 midlink(R) is magenta, PB2 CBD(R) is orange, PB2 627(R) is pink and PB2 NLS(R) is beige. FluPolB(E) core is light grey, PA ENDO(E) is light green, PB2 mid-link(E) is salmon, PB2 CBD(E) is orange, PB2 627(E) is light pink and PB2 NLS(E) is brown. **G** Cell-based split-luciferase complementation assay to assess B/Memphis/13/2003 FluPol self-oligomerisation for the indicated PA mutants. HEK-293T cells were co-transfected with plasmids encoding PB2, PA, PB1-luc1 and PB1-luc2[32]. Luminescence signals due to luciferase reconstitution are represented as a percentage of PA-WT (mean ± SD, $n = 6$, ***$p < 0.001$, one-way ANOVA; Dunnett's multiple comparisons test). Source data are provided as a Source data file. **H** Comparison of the distinct apo-FluPolB (left) and apo-FluPolA/H7N9-WT (PDB 7ZPM) (right) symmetrical dimers, with monomer 1 core (grey) in the same orientation. Other domains are coloured as in (**A**).

of a symmetrical dimer could perhaps eject the bound 5′ end (see discussion).

The cryo-EM analysis shows that most apo-FluPolB symmetric dimers exhibit a complete encapsidase conformation in one monomer (monomer 1 in Fig. 4A). In the apo-FluPolB encapsidase, the PB2 lid domain is disordered (as in FluPolA(E)), but interestingly, is observed in its normal position in the 5′ cRNA hook bound form of the encapsidase, due to subtle displacements of domains (Fig. 4A, E). For the symmetric partner monomer denoted FluPolB(S) (monomer 2 in Fig. 4A), a variety of conformations are observed including with the PA-ENDO in either the replicase (ENDO(R)), encapsidase (ENDO(E)) or transcriptase (ENDO(T)) orientations, with the PB2-C domains usually exhibiting weaker density (Supplementary Notes 4, 5). Even the core conformations vary due to different openings of the polymerase. Given that this encapsidase-containing dimer requires the hANP32A LCAR for stabilisation, we suggest that the LCAR in fact stabilises the encapsidase conformation. This is consistent with the fact that the encapsidase has only ever been visualised in the presence of ANP32, as here, or for FluPolC[22]. As described below, the FluB replication complex cryo-EM maps reveal the likely pathway of the extended LCAR, providing a rationale for how it stabilises the encapsidase conformation by electrostatic complementarity (Fig. 4F).

We used the split luciferase assay to show that mutations in the symmetrical FluPolB dimer interface significantly reduce self-oligomerisation in cells (Fig. 4G), suggesting that this dimer likely exists under physiological conditions. Furthermore, we reiterate that the FluPolA and FluPolB symmetrical dimers are quite distinct, involving different regions at their respective 2-fold interfaces (Fig. 4H).

### Structure of the influenza B replication complex
The overall architecture of the FluB replication complex is similar to that of FluA, with most domains in the equivalent location, albeit with a few significant differences (Fig. 5; Supplementary Figs. 8, 9). In the FluPolB replicase there is a rotation of ~15° of the PA-ENDO(R) towards the PB2-627(R) domain when compared to the FluA replication complex, allowing the FluPolB PA-ENDO(R) 63−73 insertion to contact the PB2-627(R) domain in the region of W575, a contact not observed in FluPolA replicase (Supplementary Fig. 8D). The encapsidase moiety of the FluB replication complex adopts a very similar conformation to that seen in the apo-FluPolB symmetrical dimer. The PA-ENDO(E) is rotated by ~48° away from the PB2-CBD(E) compared to FluPolA, so there is no longer contact with the PA-ENDO(E) 63-73 loop as observed in FluPolA encapsidase (Supplementary Fig. 8E). In addition, the lid domain of PB2(E) is also disordered in FluPolB encapsidase, although there is some suggestive, but low resolution, density close to the

PA-ENDO(E). Interestingly, there is unambiguous density for the FluPolB encapsidase PB1-C/PB2-N helical bundle swung away from its normal position (Supplementary Fig. 8E). This structural element, together with residues 194-198 at the tip of the PB1(E) β-ribbon, makes a significant new interface with the top of the PB2-CBD(R) (residues 466-474), which considerably reinforces the replicase-encapsidase interface (Fig. 5A; Supplementary Fig. 9A, F). No equivalent interaction is observed in the FluA replication complex. This extra contact largely accounts for the fact that the total buried surface between the replicase and encapsidase in the FluB replication complex is ~ 5100 Å$^2$, considerably more than the ~3300 Å$^2$ for FluA. This mainly results from the fact that the three encapsidase subunits each contact PB2(R) (Supplementary Fig. 9A–D).

Furthermore, the PA-C 609-loop, a specific FluPolB insertion that is also important for Pol II pS5 CTD binding[6], makes special contacts within the FluB replication complex. In the encapsidase, the PA-C(E) 609-loop interacts with the PB2-CBD(E) 420-loop, thereby providing additional stabilisation to the encapsidase conformation (not shown). Conversely, in the replicase, the PA-C(R) 609-loop interacts with the PA-C(E) arch domain, reinforcing the replicase-encapsidase interface (Supplementary Fig. 9E).

### Influenza B polymerase trimer
The FluB replication complex is only seen as part of a trimer (determined at 3.57 Å resolution overall), with an additional monomer, FluPolB(S), making a symmetrical FluB-type dimer interface with the replicase (Fig. 4F; Fig. 5A; Table 1; Supplementary Notes 6, 7). FluPolB(S) is less well ordered with only the core visible and not the peripheral domains, likely due to flexibility. The encapsidase component of the FluB replication complex cannot simultaneously make a symmetrical dimer as it uses the same interface (i.e. the PA-arch and PB1 β-hairpin) to interact with the replicase. This shows that the encapsidase component of the apo-FluPolB symmetrical dimer would have to disassociate to be able to form the FluB replication complex. Consistent with this and the biophysical data, a significant number of monomeric apo-encapsidases are observed (Supplementary Notes 5). A speculative biological role for the replication complex-containing FluPolB trimer is mentioned in the discussion.

### Interactions of human ANP32A within the influenza B replication complex
Human ANP32A binds in the same position and orientation to the FluPolB encapsidase as in its FluPolA counterpart, but due to sequence divergence the interactions are not necessarily conserved (Fig. 5A, B). The total buried solvent accessible surface upon hANP32A binding to

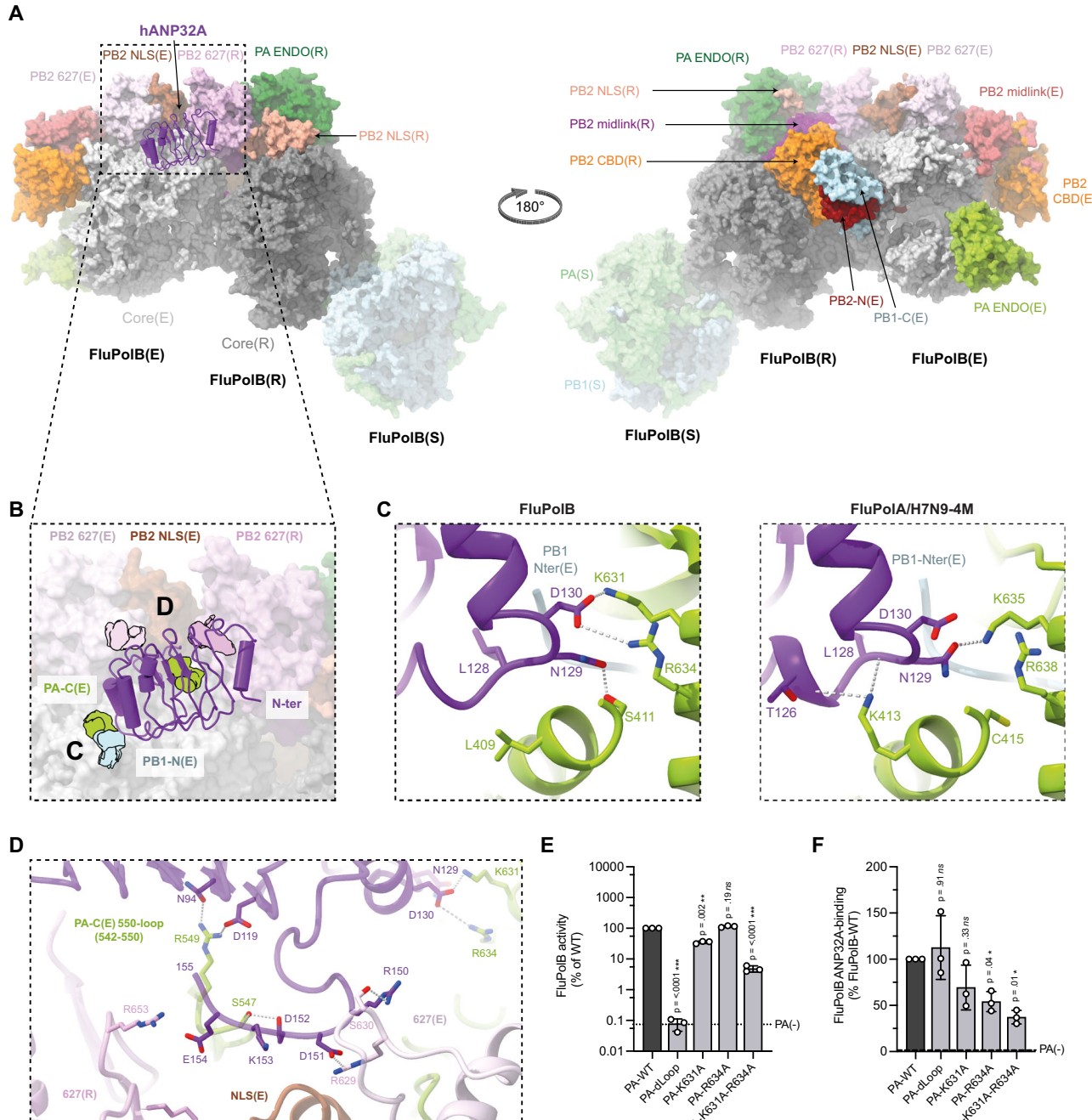

**Fig. 5 | Overall FluB trimeric replication complex and hANP32A interactions.**
**A** Surface representation of the FluB trimer replication complex. hANP32A is displayed as cartoon, coloured in purple. FluPolB replicase (R) core is coloured in dark grey. PA ENDO(R) in dark green, PB2 midlink(R) in magenta, PB2 CBD(R) in orange, PB2 627(R) in pink, PB2 NLS(R) in beige. FluPolB encapsidase (E) core is coloured in light grey, PA ENDO(E) in light green, PB2 midlink(E) in salmon, PB2 CBD(E) in orange, PB2 627(E) in light pink, PB2 NLS(E) in brown. PB1-C(E) and PB2-N(E), respectively coloured in blue and dark red, interact with PB2 CBD(R) bridging FluPolB(R) and FluPolB(E). The symmetrical FluPolB(S) is coloured as in Fig. 4.
**B** Close-up view on hANP32A interaction with FluPolB(R) and FluPolB(E). Interaction surface are highlighted, main contacts are labelled from (**C**) to (**D**), and coloured according to FluPolB interacting domains. PB1-N(E) is coloured in light blue, PA-C(E) is coloured in light green, PB2 627(E)/NLS(E), PB2 627(R) are coloured as in (**A**). **C** Comparison of the interaction between hANP32A 128-130 loop with FluPolB PA-C(E)/PB1-N(E) and FluPolA/H7N9-4M PA-C(E)/PB1-N(E). hANP32A and FluPol domains are coloured as in (**B**). Ionic and hydrogen bonding are shown as

grey dotted lines. **D** Cartoon representation of the interaction between hANP32A curved β-sheet and LRR C-terminus with FluPolB PA-C(E) 550-loop, PB2 627(E)/NLS(E), PB2 627(R). hANP32A and FluPolB domains are coloured as in (**B**). Ionic and hydrogen bonding are shown as grey dotted lines. **E** Cell-based assay of B/Memphis/13/2003 FluPol activity for the indicated PA mutants. HEK-293T cells were co-transfected with plasmids encoding PB2, PB1, PA, NP with a model vRNA encoding the Firefly luciferase. Luminescence was normalised to a transfection control and is represented as a percentage of PA-WT (mean ± SD, $n = 3$, **$p < 0.002$, ***$p < 0.001$, one-way ANOVA; Dunnett's multiple comparisons test). Source data are provided as a Source Data file. **F** Cell-based assay of B/Memphis/13/2003 FluPol binding to ANP32A for the indicated PA mutants. HEK-293T cells were co-transfected with plasmids encoding PB2, PA, PB1-luc1 and hANP32A-luc2. Luminescence signals due to luciferase reconstitution are represented as a percentage of PA-WT (mean ± SD, $n = 3$, *$p < 0.033$, one-way ANOVA; Dunnett's multiple comparisons test). Source data are provided as a Source Data file.

the FluB replication complex is ~2200 Å$^2$, only 66% of that for the FluA replication complex. In both complexes, the main anchor point remains the C-terminal end of the LRR domain wedged against the PA-C(E) domain in the vicinity of the N-terminus of PB1(E) (Fig. 5A, B). On the other hand, in the FluB replication complex, the N-terminal end of the LRR domain does not make any contacts and consequently has less well-ordered density due to mobility. A critical contact is again made by hANP32A 129-ND, but in the FluB replication complex it is D130 that directly interacts with the Pol II pS5 CTD binding residue PA(E)/K631 (in FluA replication complex, N129 contacts the equivalent PA(E)/K635), with also a slightly more distant salt-bridge to R634, whereas N129 hydrogen bonds to PA(E)/S411 (Fig. 5C). The equivalent of FluPolA PA/K413 is L409 in FluPolB and is close but does not interact (Fig. 5C). It has previously been shown that the substitution N129E that occurs in hANP32E is responsible for its limited ability to support FluB replication[14], emphasising the importance of this contact point. Furthermore, we show that the double mutation PA/K631A and R634A reduces FluPolB activity (Fig. 5E), as previously described[6], as well as hANP32A binding to FluPolB in a cell-based split luciferase assay (Fig. 5F). In FluPolA, PA(E)/R638 is further away and its mutation less impacts replication[3]. Furthermore, the PA-C(E) 550-loop of FluPolB, being four residues shorter than that of FluPolA, does not reach so far onto the β-sheet surface of hANP32A (Fig. 5B, D). This observation is consistent with the fact that no decreased binding of the FluPolB PA-C 550-loop deletion mutant to hANP32A was observed (Fig. 5F), despite some interactions between PA(E)/S547 to hANP32A/D152 and the carbonyl oxygen of K153, and PA(E)/R549 that contacts hANP32A/N84 and D119. Other polar interactions are made to hANP32A/R150 and D151 by PB2(E)/S630 and R629, respectively, and to hANP32A/E154 by PB2(R)/R653 and possibly K639 (Fig. 5D).

### Path of the human ANP32A LCAR in the influenza B replication complex

Additional pseudo-continuous density is present in the FluB replication complex maps that we interpret as tracing the path of the hANP32A LCAR extending beyond residue 155, the last of the folded part of the LRR domain, and wrapping around the encapsidase (Fig. 6A, B). Although no accurate model can be built into this density, it could correspond to a chain of at least 50 residues extending to beyond hANP32A residue 200. The path follows a clearly defined positively-charged electrostatic track, created by a number of basic residues that would point towards the acidic LCAR. In order along the pathway, we find: PB2-NLS(E)/K721, K742, R741, K703, K734 and PB2-627(R)/R629, K588; PB1(E)/K566, K570 and PA(E)/K298, K301, K475, H506, K374; PB1(E)/R196 and R203 that are close to the PB1 NLS on the long β-ribbon[47], and finally PB1(E)/R135, K353 and PB2(E)/R40 (Fig. 6C). The LCAR passes over the PB2-NLS(E)/PB2-627(R) interface, then over the 3′ end secondary binding loop (PA/K298, K301), round the tip of the PB1(E) β-ribbon (PB1/R203) and then parallel to the β-ribbon with the PA(E) arch on the other side (PA/K374) (Fig. 6B, C). These observations plausibly explain how the LCAR stabilises the encapsidase conformation by electrostatic complementation.

### Model of the active RNA bound replication complex

We have determined the structure of the putative FluA and FluB replication complexes in the absence of viral RNA. We therefore sought to model how template and product RNA could bind to an active replication complex (Fig. 6D–F). To remain simple, we have chosen not to take into account the expected conformational changes that are known to accompany promoter binding and the initiation to elongation transition, which involves FluPol core opening and extrusion of the priming loop[48,49]. The template extremities and product-template duplex bound to the replicase were modelled by superposing on PB1(R) the FluPolA/H7N9 elongation state (PDB 8PNQ) with the 3′ end of the template binding back to the secondary site. The replication

product was manually extended from the top of the duplex through a channel into the 5′ end hook binding site of the encapsidase, modelled using the cRNA 5′ hook-bound FluPolB encapsidase structure. From the 3′ end at the +1 position in the replicase active site to the 5′ end in the encapsidase hook-binding site, a minimal 32 nucleotides of the product are modelled (10 in the duplex, 12 in the channel, 10 in the 5′ hook) (Fig. 6D, E). In the FluA replication complex, the putative product exit channel linking replicase to encapsidase passes between the PB2-N(R) and PB2-CBD(R) domains and then has PA-C(E) on one side, while the other side is solvent exposed and could be where the extending product bulges out to be bound by incoming NP (Fig. 6E, F). The channel is lined by basic residues notably PB2(R)/R143, R144, K157, R213, R368, R369, K389 and PA-C(E)/K361, K362, K367, R508 yielding a positively charged pathway rich in flexible arginines that are able to interact with both RNA bases and phosphate backbone (Fig. 6E, F). Not unexpectedly, the product exit channel partially overlaps with the capped transcription primer entrance channel, where it has already been shown how some of the same arginines can adapt to interact with different RNA configurations[27]. The RNA model is transferable between the FluA and FluB replication complexes without modification. In the FluB complex, the distal, C-terminal end of the modelled LCAR, would clash with the product RNA entering the encapsidase hook-binding site (Fig. 6D, F). Consequently, the LCAR would have to be displaced as the product RNA emerges. This could have the dual effect of preventing non-specific RNA binding to the encapsidase prior to product emergence, with subsequent release of the LCAR to be able to recruit NP to the elongating replication product.

## Discussion

De novo synthesis of the influenza anti-genomic and genomic RNA (cRNA and vRNA), respectively from parental vRNPs and cRNPs, with concomitant packaging of the product RNA into a progeny RNP, is a highly complex process that we are only just beginning to get a grasp off. Two key elements have been established, firstly that the host factor ANP32 plays critical and probably multiple roles in the process and secondly, that an asymmetric FluPol dimer comprising replicase (integrated into the parental RNP) and encapsidase (a newly synthesised and initially apo-FluPol) is fundamental to nucleate co-replicational assembly of the progeny RNP. The proposed roles of ANP32 so far include formation and stabilisation of the ternary replicase-ANP32-encapsidase replication complex[22] and secondly, through interactions of the LCAR with apo-NP, successive recruitment of NPs to package the growing replicate[18,19]. A further intriguing aspect that lacks a full mechanistic explanation is why certain specific mutations, mainly in PB2, are required to overcome the restriction of avian strain FluPols to replicate in human cells, as this mainly depends on a 33 residue insertion in the LCAR of avANP32A, compared to hANP32A and hANP32B[8].

Here we present evidence based on in vitro biochemical and structural analysis of complexes of FluPolA and FluPolB with hANP32A, that ANP32 may have a third important role that is to act as an electrostatic chaperone/disaggregase[24] for apo-FluPol. This function is primarily dependent on the LCAR and involves solubilisation and stabilisation of apo-FluPolA and apo-FluPolB predominantly in a dimeric form at physiological salt concentrations. We speculate that this may have been the primordial role of ANP32 in the nuclear lifecycle of influenza-like viral polymerases, since it is highly conserved in most eukaryotes, notably those known to be hosts of orthomyxo- and orthomyxo-like viruses. Only later, ANP32 would have acquired an active role in replication.

Interestingly, both FluPolA and FluPolB apo-dimers have 2-fold symmetric core interfaces (although the peripheral domains need not be disposed symmetrically), but they are structurally quite different (Fig. 4H). The FluPolA apo-dimer is stable at high salt without ANP32[26,27] but ANP32 binding at physiological salt enhances the dimer

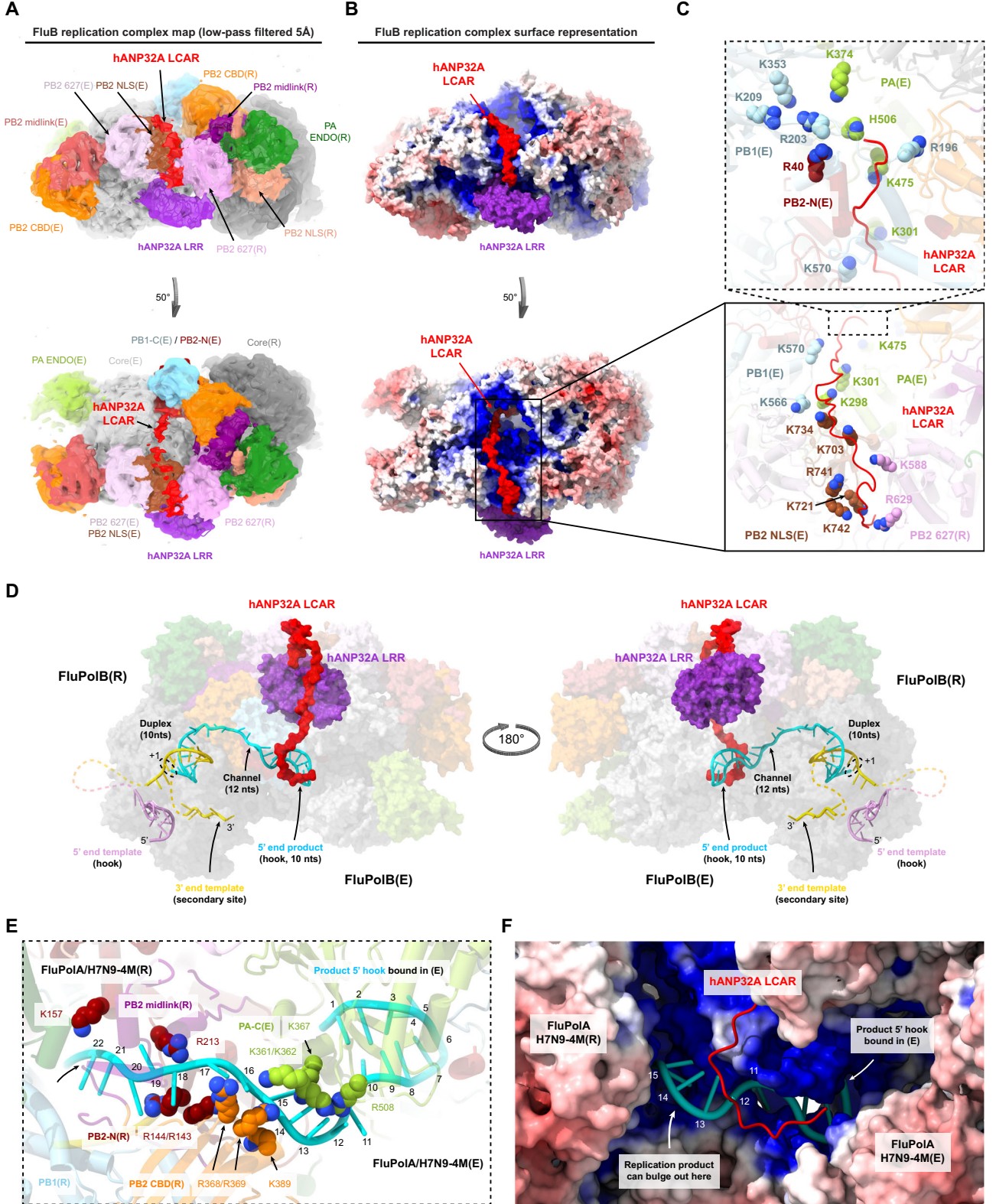

population (Supplementary Fig. 2). On the other hand, FluPolB is monomeric at high salt (Fig. 1), but again ANP32 significantly increases the amount of dimers at lower salt. One monomer of the FluPolB apo-dimer is observed to be preferentially in the encapsidase conformation, which appears specifically to be stabilised by the LCAR of hANP32A. These initial observations led us to perform cryo-EM structural analysis on mixtures of apo-FluPolA or apo-FluPolB, with hANP32A, which yielded multiple structures including the replication

complex of each FluPol. Electrostatic calculations for the FluA replication complex structure suggest that human adaptive mutations restore a coherent positively charged pathway at the PB2-NLS(E)/PB2-627(R) interface, able to bind the negatively charged proximal region of the LCAR of hANP32A or hANP32B. In contrast, the more mixed surface of avian FluPol more appropriately binds the avian LCAR, which because of the 33 residue insertion, has more basic and hydrophobic residues in the equivalent region (Fig. 3). The FluA replication

**Fig. 6 | hANP32A LCAR density and putative RNA path within FluA/H7N9-4M and FluB replication complexes. A** Low-pass filtered FluB replication complex map at 5 Å resolution (threshold 0.12). hANP32A and FluPolB domains are coloured as in Fig. 5. Density assigned to the hANP32A LCAR (red) passes over the NLS(E) and 627(R) interface (top) and then extends over FluPolB(E) (50 degree rotated view, bottom). **B** As for (**A**), but with the FluB replication complex surface coloured by electrostatic potential. The hANP32A LCAR follows a positively charged pathway. **C** Close-up view showing the residues forming the positively-charged track followed by hANP32A LCAR (red cartoon). Domains (cartoon) and residues (spheres) are coloured as in (**A**). **D** Model of template and product RNA binding to the FluB replication complex in the early-elongation state shown in two orientations as a transparent surface. hANP32A LRR and LCAR are respectively coloured purple and red. The template 5′ hook is in plum and the 3′ end gold. The replication product is coloured cyan. The template extremities and product-template duplex bound in the replicase core are modelled from PDB 8PNQ. The replication product is modelled to extend through a channel between the replicase PB2-N and midlink/cap-binding domains into the 5′ end hook binding site of the encapsidase (modelled using the cRNA hook-bound FluPolB(E) structure). The template 3′ end binds in the

replicase secondary site. Un-modelled RNA is displayed as dotted lines. The C-terminal end of the modelled LCAR (**A**–**C**) would clash with the product RNA entering the encapsidase hook binding site and would have to be displaced during elongation. **E** Close-up view of the elongating RNA product, coloured as in (**D**), exiting the replicase and entering the encapsidase, mapped onto the FluA/H7N9-4M replication complex. Nucleotides 1–22 are numbered from the 5′ end of the product, with nucleotides 1–10 corresponding to the 5′ hook bound to the encapsidase and 11–22 to the minimal product in the channel formed by PB2-N/midlink/CBD(R) and PA-C(E). Basic residues in the channel are displayed with atoms as spheres. Domains are coloured as in (**A**), with PB1(R) in blue, PB2 midlink(R) in magenta, PB2-N(R) in dark red, PB2 CBD(R) in orange and PA-C(E) in green. **F** Surface representation of the FluA/H7N9-4M replication complex coloured according to the electrostatic potential, with the modelled RNA product and hANP32A LCAR superimposed, numbered and coloured as in (**E**). The distal end of the hANP32A LCAR clashes with the RNA product. White arrows indicate the 5′ hook bound to the encapsidase and where the growing replication product could bulge out to be bound by NP.

complex structure also explains why PB2/E627 is only restrictive in human cells in the replicase. The FluB replication complex exhibits density that we assign to the extended LCAR following a basic pathway around the encapsidase, thus providing a plausible explanation of how it stabilises this particular FluPol conformation (Fig. 6A–C). Both the FluA and FluB replication complexes also have a positively-charge channel for the product RNA to exit the replicase and enter the 5′ hook binding site of the encapsidase (Fig. 6D–F).

Whether or not the chaperone effect of ANP32 is important in vivo, it is clear that hANP32A binds to dimeric apo-FluPol in vitro (Fig. 1) and in cells (e.g. Supplementary Fig. 4E, J, for split luciferase binding assays in the absence of viral RNA). This likely occurs in the nucleus prior to the apo-FluPol encountering an RNP. Thus, for both FluA and FluB, an ANP32-bound apo-FluPol symmetric dimer is brought together with a replication-competent RNP during the replication initiation process. We therefore propose a generalised trimer model of replication. For both FluPolA (Supplementary Fig. 10) and FluPolB (Supplementary Fig. 11), one half of the dimer would become the ANP32-bound encapsidase within the asymmetric replication complex. For FluPolA, the third polymerase has been proposed to play a role in template realignment specifically during initiation of cRNA to vRNA synthesis by forming a transient symmetrical interface with the asymmetric replication complex dimer[26]. However, such a trimer has not been observed structurally. For FluPolB, based on the observed trimer structure, where the third FluPolB forms a different symmetrical interface with the replicase, we speculate that it could have a distinct role. It might assist in replication termination by binding the replicase and releasing the 5′ hook of the template so it can be copied. It is possible that both types of symmetric dimer exist for both FluPolA and FluPolB under certain conditions, but this has not been shown yet. However, given that FluPolA mutated to be monomeric appears to be functional[7,15], it could also be that these third polymerases are not essential but merely increase efficiency of replication.

To reveal more details of the mechanism, it will be important to determine structural snapshots of replication in action, as has been done for transcription[48,49]. This would visualise the trajectory of the nascent RNA replicate from the replicase into the encapsidase and subsequent conformational changes that might occur as elongation proceeds. Ultimately, one would like to validate the NP recruitment model leading to RNP assembly.

Finally, our structures will be useful in interpreting polymerase mutations detected in mammals infected with highly pathogenic and potentially pandemic avian influenza strains (as exemplified by the ongoing outbreaks of H5N1 in diverse mammals, notably cows), for instance, whether they might be mammalian adaptive mutations that effect the stability of the replication complex[28,29].

## Methods

### Construction of expression plasmids for influenza A/Zhejiang/DTID-ZJU01/2013(H7N9) and B/Memphis/13/2003 polymerases

The two previously described pFastBac Dual vector encoding for the influenza A/Zhejiang/DTID-ZJU01/2013(H7N9) wild-type polymerase subunits, PA (Uniprot: M9TI86), PB1 (Uniprot: M9TLW3) and PB2 (Uniprot: X5F427)[48] and the A/Zhejiang/DTID-ZJU01/2013(H7N9) 4 M monomeric mutant bearing the mutations PA/E349K, R490I, PB1/K577G and PB2/G74R[7] were used as a starting point. The PA N-terminal His-tag was removed by a combination of PCRs and Gibson assembly. The PB2 C-terminal Twin-strep-tag was kept.

The previously described pKL vector encoding the self-cleavable poly-protein of influenza B/Memphis/13/2003 polymerase subunits, PA (Uniprot: Q5V8Z9_9INFB), PB1 (Uniprot: Q5V8Y6_9INFB) and PB2 (Uniprot: Q5V8X3_9INFB)[50] was used as a starting point. Each FluPolB subunit was amplified and inserted in a pLIB plasmid by a combination of PCRs and Gibson assembly resulting in FluPolB subunits being under control of distinct polyhedrin promoters. The PA N-terminal His-tag was removed, whilst the PB2 C-terminal Twin-strep-tag was retained to enable purification.

All plasmid sequences were confirmed by Sanger sequencing for each polymerase subunit.

Removal of all N- or C-terminal tags (except the PB2 C-terminal purification tag) was essential to allow replication complex formation. An N-terminal PA tag prevents formation of the full replicase conformation, which requires close contact of the cap-binding domain to the N-terminus of PA. An N-terminal PB1 extension impedes close packing against PA-C of ANP32 in the vicinity of residues 129-130, which buries the PB1 N-terminus. In the FluB replication complex, an N-terminal PB2 extension would prevent the observed contact between the encapsidase PB1-C/PB2-N helical bundle with the replicase cap-binding domain.

### Influenza A and B polymerases expression and purification

FluPolA/H7N9-WT, FluPolA/H7N9-4M and FluPolB were produced using the baculovirus expression system in *Trichoplusia ni* High 5 cells. For large-scale expression, cells at $0.8 \cdot 10^6$ cells/mL concentration were infected by adding 1% of virus. Expression was stopped 72 to 96 h after the day of proliferation arrest and cells were harvested by centrifugation (1000 g, 20 min at 4 °C). Cells were disrupted by sonication for 5 min (5 s ON, 20 s OFF, 40% amplitude) on ice in lysis buffer (50 mM HEPES pH 8, 500 mM NaCl, 2 mM TCEP, 5% glycerol) with cOmplete EDTA-free Protease Inhibitor Cocktail (Roche). After lysate centrifugation at 48,000 g for 45 min at 4 °C, ammonium sulphate was added to the supernatant at 0.5 g/mL final concentration. The recombinant protein was then collected by centrifugation (45 min,

4 °C at 70.000 $g$), re-suspended in the lysis buffer, and the procedure was repeated. FluPol was purified using strep-tactin affinity purification beads (IBA, Superflow). Bound proteins were eluted using the lysis buffer supplemented by 2.5 mM d-desthiobiotin and protein-containing fractions were pooled and diluted with an equal volume of buffer (50 mM HEPES pH 8, 2 mM TCEP, 5% glycerol) before loading on an affinity column HiTrap Heparin HP 5 mL (Cytiva). A continuous gradient of lysis buffer supplemented with 1 M NaCl was applied over 15 CV, and FluPol was eluted as single species at ~800 mM NaCl. Pure and acid nucleic free FluPol were dialysed overnight in a final buffer (50 mM HEPES pH 8, 500 mM NaCl, 2 mM TCEP, 5% glycerol), concentrated with Amicon Ultra-15 (50 kDa cutoff), flash-frozen and stored at −80 °C for later use.

## Human Acidic Nuclear Phosphoprotein 32 A

Human ANP32A (hANP32A) was cloned and expressed as previously described[7]. The N-terminal His-tagged hANP32A construct was expressed in BL21(DE3) *E.coli* cells. Expression was induced with 1 mM IPTG, for 4 h at 37 °C. Cells were harvested by centrifugation (1000 $g$, 20 min at 4 °C), disrupted by sonication for 5 min (5 s ON, 15 s OFF, 50% amplitude) on ice in lysis buffer (50 mM HEPES pH 8, 150 mM NaCl, 5 mM β-mercaptoethanol (BME) with cOmplete EDTA-free Protease Inhibitor Cocktail (Roche). After lysate centrifugation at 48,000 $g$ for 45 min at 4 °C, the soluble fraction was loaded on a HisTrap HP 5 mL column (Cytiva). Bound proteins were subjected to a wash step using the lysis buffer supplemented by 50 mM imidazole. Remaining bound protein was eluted using the lysis buffer supplemented by 500 mM imidazole. Fractions containing hANP32A were dialysed overnight in the lysis buffer (50 mM HEPES pH 8, 150 mM NaCl, 5 mM BME) together with N-terminal his-tagged TEV protease (ratio 1:5 w/w). Tag-cleaved hANP32A protein was subjected to a Ni-sepharose affinity column to remove the TEV protease, further concentrated with Amicon Ultra-15 (3 kDa cutoff) and subjected to a size-exclusion chromatography (SEC) using a Superdex 200 Increase 10/300 GL column (Cytiva) in a final buffer containing 50 mM HEPES pH 8, 150 mM NaCl, 2 mM TCEP. Fractions containing exclusively hANP32A were concentrated with Amicon Ultra-15 (3 kDa cutoff), flash-frozen and stored at −80 °C for later use.

Truncated hANP32A constructs (1–199 and 144–249) were generated, expressed and purified as previously described[21]. The hANP32A 1–149 construct was a gift from Cynthia Wolberger (Addgene plasmid # 67241[51]), and was expressed and purified as previously described[21].

## Analytical size exclusion chromatography

SEC experiments for FluPolA/H7N9-4M and FluPolB were performed on a Superdex 200 Increase 3.2/300 (Cytiva) at 4 °C, in a final buffer containing 50 mM HEPES pH 8, 500-300-150-100 mM NaCl, 2 mM TCEP. Depending on the experiment, 5 μM FluPol, 15 μM hANP32A (full-length, '1–149', '1–199', '144-Cter') and 10 μM 5' vRNA 1-12 (5'-pAGU AGU AAC AAG-3') were used. Resulting mixtures were incubated 1 h on ice and centrifuged 5 min at 11,000 $g$ prior to injection onto the column. SEC fractions of interest were loaded on 4–20% Tris-glycine gel (ThermoFisher) and stained with Coomassie Blue.

SEC experiments for FluPolA/H7N9-WT were performed on a Superdex 200 Increase 3.2/300 (Cytiva) at 4 °C, in a final buffer containing 50 mM HEPES pH 8, 500-300-150 mM NaCl, 2 mM TCEP. Depending on the experiment, 5 μM FluPol, 15 μM hANP32A and 10 μM 5' vRNA 1-14 (5'-pAGU AGU AAC AAG AG)/3' 1-18 (5'-UAU ACC UCU GCU UCU GCU -3') were used. Resulting mixtures were incubated 1 h on ice and centrifuged 5 min at 11,000 $g$ before injection onto the column. SEC fractions of interest were loaded on 4–20% Tris-glycine gel (ThermoFisher) and stained with Coomassie Blue.

Source data are provided as a Source Data file.

## Mass photometry analysis

Mass photometry measurements were performed on a OneMP mass photometer (Refeyn). Coverslips (No. 1.5H, 24 × 50 mm, VWR) were washed with water and isopropanol before being used as a support for silicone gaskets (CultureWellTM 423 Reusable Gaskets, Grace Bio-labs). Contrast/mass calibration was realised using native marker (Native Marker unstained protein 426 standard, LC0725, Life Technologies) with a medium field of view and monitored during 60 s using the AcquireMP software (Refeyn). For each condition, 18 μl of buffer (50 mM HEPES pH 8, 100/150/300/500 mM NaCl, 2 mM TCEP) were used to find the focus. Using diluted SEC inputs, 2 μl of sample were added to reach a final FluPol concentration of 50 nM. Movies of 60 s were recorded, processed and mass estimation was determined automatically using the DiscoverMP software (Refeyn).

Source data are provided as a Source Data file.

## Electron microscopy

**FluPol A/Zhejiang/DTID-ZJU01/2013(H7N9) and B/Memphis/13/ 2003 replication complexes sample preparation.** FluPolA/H7N9-4M and FluPolB replication complexes were trapped by mixing 1.15 μM FluPol with 5.75 μM hANP32A (molar ratio 1:5) in a final buffer containing 50 mM HEPES pH 8, 150 mM NaCl, 2 mM TCEP. Mix were incubated for 1 h at 4 °C, centrifuged for 5 min at 11,000 $g$ and kept at 4 °C before proceeding to grids freezing. For grid preparation, 1.5 μl of sample was applied on each sides of plasma cleaned (Fischione 1070 Plasma Cleaner: 1 min 10 s, 90% oxygen, 10% argon) grids (UltrAufoil 1.2/1.3, Au 300). Excess solution was blotted for 3 s, blot force 0, 100% humidity, at 10 °C, with a Vitrobot Mark IV (ThermoFisher) before plunge freezing in liquid ethane.

**FluPol B/Memphis/13/2003 bound to 5' cRNA sample preparation.** The FluPolB encapsidase bound to 5' cRNA structure was trapped by mixing 1.15 μM FluPolB with 5.75 μM hANP32A and 1.72 μM 5' cRNA 1-12 (5'-AGC AGA AGC AGA -3') (molar ratio 1:5:1.5) in a final buffer containing 50 mM HEPES pH 8, 150 mM NaCl, 2 mM TCEP. The mix was incubated for 1 h at 4 °C, centrifuged for 5 min at 11,000 $g$ and kept at 4 °C before proceeding to grid freezing. For grid preparation, 1.5 μl of sample was applied on each sides of plasma cleaned (Fischione 1070 Plasma Cleaner: 1 min 10 s, 90% oxygen, 10% argon) grids (UltrAufoil 1.2/1.3, Au 300). Excess solution was blotted for 3 s, blot force 0, 100% humidity, at 10 °C, with a Vitrobot Mark IV (ThermoFisher) before plunge freezing in liquid ethane.

## Cryo-EM data collection

**FluPol A/Zhejiang/DTID-ZJU01/2013(H7N9) and B/Memphis/13/ 2003 replication complexes.** Automated data collections were performed on a TEM Titan Krios G3 (ThermoFisher) operated at 300 kV equipped with a K3 direct electron detector camera (Gatan) and a BioQuantum energy filter (Gatan), using EPU (ThermoFisher). Coma and astigmatism correction were performed on a carbon grid. Micrographs were recorded in counting mode at a ×105,000 magnification giving a pixel size of 0.84 Å with defocus ranging from −0.8 to −2.0 μm. Gain-normalised movies of 40 frames were collected with a total exposure of ~40 $e^-/Å^2$.

**FluPol B/Memphis/13/2003 bound to 5' cRNA sample preparation.** Automated data collection was performed on a TEM Glacios (ThermoFisher) operated at 200 kV equipped with a F4i direct electron detector camera (ThermoFisher) and a SelectrisX energy filter (ThermoFisher), using EPU (ThermoFisher). Coma and astigmatism correction were performed on a carbon grid. Micrographs were recorded in counting mode at a ×130,000 magnification giving a pixel size of 0.878 Å with defocus ranging from −0.8 to −2.0 μm. EER movies were collected with a total exposure of ~40 $e^-/Å^2$.

## Image processing

**FluPol A/Zhejiang/DTID-ZJU01/2013(H7N9) structure determination.** For the FluPolA TEM Titan Krios dataset, 14,001 movies were collected. Movie drift correction was performed using Relion's Motioncor implementation, with 7 × 5 patch, using all movie frames[52]. All additional initial image processing steps were performed in cryoSPARC v4.3[53]. CTF parameters were determined using 'Patch CTF estimation'. Realigned micrographs were then manually inspected and low-quality images were manually discarded resulting in 13,328 micrographs kept. Particles were automatically picked using a circular blob with a diameter ranging from 110 to 130 Å, and extracted using a box size of 420 × 420 pixels$^2$, Fourier cropped to 210 × 210 pixels$^2$. Successive 2D classifications were used to eliminate particles displaying poor structural features, and coarsely separate monomers from dimers. Monomers were subjected to a 'heterogeneous refinement' job. Particles displaying PA-ENDO in the replicase conformation (PA-ENDO(R)), the rest of them displaying a dislocated FluPol core, were Fourier uncropped and subjected to a 'non-uniform refinement' job. Based on the estimated particle angles and shifts, a '3D classification' job was performed. For each relevant FluPol conformation, particles were grouped and subjected to a final 'non-uniform refinement'. FluPolA/H7N9-4M asymmetric dimers were first subjected to a 'heterogeneous refinement' job. Particles assigned to the 3D class displaying well-defined secondary structures were used for model training and picking using Topaz[54]. The resulting picked particles were extracted and subjected to 2D classification. All asymmetric dimers particles were merged, the duplicates removed, Fourier uncropped and then subjected to a 'non-uniform refinement' job. To alleviate the preferential orientation problem of the FluPolA/H7N9-4M replication complex, a '3D classification' job was used. Particles displaying a proper view distribution equilibrium were used and subjected to a 'non-uniform refinement'. Based on this consensus map, particle subtraction around 'FluPol(R) minus 627(R)' and 'FluPol(E)-hANP32A-627(R)' was performed. The subtracted particles were finally subjected to local refinement to improve subtracted particle angles and shifts estimation. Post-processing was performed in cryoSPARC using an automatically or manually determined B-factor. For each final map, reported global resolution is based on the FSC 0.143 cut-off criteria. Local resolution variations were estimated in cryoSPARC. The 3D-FSCs and particle orientation distribution were calculated in cryoSPARC v4.4.1, using the 'Orientation diagnostics' job. The detailed image processing pipeline is shown in Supplementary Notes 1–3.

**FluPol B/Memphis/13/2003 structure determination.** For the FluPolB TEM Titan Krios dataset, 15,650 movies were collected. Movie drift correction was performed using Relion's Motioncor implementation, with 7 × 5 patch, using all movie frames[52]. All additional initial image processing steps were performed in cryoSPARC v4.3[53]. CTF parameters were determined using 'Patch CTF estimation', realigned micrographs were then manually inspected and low-quality images were manually discarded resulting in 15,234 micrographs kept. Particles were automatically picked using a circular blob with a diameter ranging from 110 to 140 Å and extracted using a box size of 480 × 480 pixels$^2$, Fourier cropped to 200 × 200 pixels$^2$. Successive 2D classifications using a circular mask of 210 Å were used to eliminate particles displaying poor structural features. Following initial 2D classifications, all particles were re-extracted at a larger box size (512 × 512 pixels$^2$, Fourier cropped to 200 × 200 pixels$^2$) and subjected to multiple 2D classifications using a circular mask of 280 Å to coarsely separate dimers, monomers and trimers. For the FluPolB symmetrical dimers, following an 'ab-initio' reconstruction job, particles displaying one FluPolB(E) were Fourier uncropped and subjected to a 'non-uniform refinement' job, followed by respective FluPolB symmetrical (FluPolB(S)) and FluPolB(E) signal subtraction. After subsequent local refinements, '3D classification' jobs were performed to separate the different FluPolB states. 3D classes displaying a complete FluPol(E) conformation were grouped, locally refined and subjected to a final 'non-uniform refinement' using the un-subtracted particles. A similar approach was used for the different FluPol(S) conformations (core, ENDO(R), ENDO(E) or ENDO(T)) (Supplementary Notes 4). Dimers displaying two FluPolB core were Fourier uncropped and subjected to a 'non-uniform refinement' job followed by a '3D classification'. Particles displaying one FluPolB with PA-ENDO in a transcriptase conformation (ENDO(T)) were grouped and subjected to a final 'non-uniform refinement' job (Supplementary Notes 5). For the FluPolB monomers, particles were subjected to an 'ab-initio' reconstruction followed by a 'non-uniform refinement'. Subsequent '3D classification' allowed isolation of monomeric apo-FluPolB(E). Particles were Fourier uncropped and subjected to a final 'non-uniform refinement' job (Supplementary Notes 5). For the FluPolB trimers (FluPolB replication complex plus one FluPol(S)), particles were subjected to an 'ab-initio' reconstruction job. The few particles displaying a well-defined FluB replication complex were Fourier uncropped and subjected to a 'non-uniform refinement' job. Particle subtraction was performed on 'FluPolB(S) + FluPolB(E)', 'FluPolB(S) + FluPolB(R)' and 'FluPolB(S)' moieties, followed by local refinements to improve subtracted particle angles and shifts estimation (Supplementary Notes 6). Post-processing was performed in cryoSPARC using an automatically or manually determined B-factor. For each final map, reported global resolution is based on the FSC 0.143 cut-off criteria. Local resolution variations were estimated in cryoSPARC. The 3D-FSCs and particle orientation distribution were calculated in cryoSPARC v4.4.1, using the 'Orientation diagnostics' job. The detailed image processing pipeline is shown in Supplementary Notes 4–7.

**5' cRNA bound FluPol B/Memphis/13/2003 structure determination.** For the TEM Glacios dataset, 2,451 movies were collected. Movie drift correction was performed using Relion's Motioncor implementation, with 5 × 5 patch, using all movie frames[52]. All additional initial image processing steps were performed in cryoSPARC v4.3[53]. CTF parameters were determined using 'Patch CTF estimation', realigned micrographs were then manually inspected and low-quality images were manually discarded resulting in 2353 micrographs kept. Particles were automatically picked using a circular blob with a diameter ranging from 110 to 140 Å and extracted using a box size of 380 × 380 pixels$^2$, Fourier cropped to 240 × 240 pixels$^2$. Successive 2D classifications using a circular mask of 210 Å were used to eliminate particles displaying poor structural features. Remaining particles were subjected to a 'heterogeneous refinement' job. Particles belonging to the class corresponding to 5' cRNA bound FluPolB(E) were subjected to a 'non-uniform refinement' job, followed by '3D classification'. A final 'non-uniform refinement' has been done with particles displaying a complete FluPol(E) conformation. Post-processing was performed in cryoSPARC using an automatically determined B-factor. For each final map, reported global resolution is based on the FSC 0.143 cut-off criteria. Local resolution variations were estimated in cryoSPARC. The detailed image processing pipeline is shown in Supplementary Notes 9.

## Model building and refinement

Atomic models were constructed by iterative rounds of manual model building with COOT[55] and real-space refinement using Phenix, with Ramachandran restraints[56]. For model building of the replicase-moiety of the FluPolB replication complex, the previously determined replicase-like structure (PDB: 5EPI)[34] was used as starting point. The FluPolB encapsidase conformation was initially constructed from the higher resolution symmetrical dimer map and transferred to the replicase complex. For the FluA replication complex structure building, a variety of previous A/H7N9 structures were used as starting models.

Validation was performed using Phenix. Model resolution according to the cryo-EM map was estimated at the 0.5 FSC cutoff. Structural analysis was performed in Coot and Chimera[57]. Electrostatic potential surfaces were calculated using the APBS-PDB2PQR software suite[58]. Buried solvent accessible surfaces were calculated using PISA[59] at the PDBe. Figures were generated using ChimeraX[60].

## Cells

HEK-293T cells (ATCC CRL-3216) and HEK-293T (ATCC CRL-11268) ANP32AB KO cells[7] were grown in complete Dulbecco's modified Eagle's medium (DMEM, Gibco) supplemented with 10% foetal bovine serum (FBS) and 1% penicillin-streptomycin (Gibco). Cell cultures were PCR-tested regularly to ensure absence of mycoplasma contamination. Cells (3E04/well) were seeded in 96-well white plates (Greiner Bio-One) the day before transfection with polyethyleneimine (PEI-max, #24765-1 Polysciences Inc).

## Plasmids used in cell-based assays

The pcDNA3.1-hANP32A-FLAG, A/WSN/33 (WSN) pcDNA3.1-PB2, -PB1, -PA, pCI-NP and B/Memphis/13/2003 (Memphis) pcDNA3.1-PB2, -PB1, -PA, -NP plasmids were described previously[3,19,50]. Plasmids used for vRNP reconstitution assays and the WSN pCI-PB1-luc1, Memphis pCI-PB1-luc1, pCI-hANP32A-luc2, pCI-chANP32A-luc2 plasmids used for split-luciferase-based complementation assays were described previously[6,7]. The pCI-hANP32B-luc2 plasmid was constructed by replacing the hANP32A sequence in the pCI-hANP32A-luc2 plasmid. pcDNA3.1-hANP32B-FLAG, -chANP32A-FLAG were constructed by replacing the hANP32A sequence in the pcDNA3.1-hANP32A-FLAG plasmid. All mutations were introduced by an adapted QuikChange site-directed mutagenesis (Agilent Technologies) protocol[61]. ORFs were verified by Sanger sequencing.

## vRNP reconstitution assays

HEK-293T cells were co-transfected with plasmids encoding the vRNP protein components (PB2, PB1, PA, NP), a pPolI-Firefly plasmid encoding a negative-sense viral-like RNA expressing the Firefly luciferase and the pTK-Renilla plasmid (Promega) as an internal control. For FluPol activity rescue experiments in ANP32AB KO cells, a plasmid encoding either the wild-type or mutant hANP32A, hANP32B or chANP32A protein was co-transfected. Mean relative light units (RLUs) produced by the Firefly and Renilla luciferase, reflecting the viral polymerase activity and transfection efficiency, respectively, were measured using the Dual-Glo Luciferase Assay System (Promega) on a Centro XS LB960 microplate luminometer (Berthold Technologies, MikroWin Version 4.41) at 48 hours post-transfection (hpt). Firefly luciferase signals were normalised with respect to Renilla luciferase signals. At least three independent experiments (each in technical duplicates) were performed, and each biological replicate is represented as a dot in the graphs. Plasmid combinations, orientations of tags as well as plasmid amounts used for transfections in a given experiment are available as a Source data file.

## Protein complementation assays

HEK-293T cells were co-transfected with plasmids encoding the FluPol subunits (PB2, PB1-luc1, PA) and an ANP32A protein (hANP23A-luc2, hANP32B-luc2 or chANP32A-luc2). Cells were lysed 20–24 hpt in Renilla lysis buffer (Promega) for 45 min at room temperature under steady shaking. RLUs produced by the reconstituted Gaussia princeps luciferase, reflecting the FluPol-ANP32 interaction, were measured on a Centro XS LB960 microplate luminometer (Berthold Technologies, MikroWin Version 4.41) using a reading time of 10 s upon injection of 50 µl Renilla luciferase reagent (Promega). Three independent experiments (each in technical triplicates) were performed, and each biological replicate is represented as a dot in the graphs. Plasmid combinations, orientations of tags as well as plasmid amounts used for transfections in a given experiment are available as a Source data file.

## Antibodies and immunoblots

Total cell lysates were prepared in RIPA cell lysis buffer as described previously[62]. Proteins were separated by SDS-PAGE using NuPAGE™ 4–12% Bis-Tris gels (Invitrogen) and transferred to nitrocellulose membranes which were incubated with primary antibodies directed against PA (1:2500[63]), PB2 (GTX125925, GeneTex, 1:5000), Gaussia princeps luciferase (New England Biolabs, #E8023, 1:5000), Histone H3 (Cell Signalling Technology, #9715, 1:1000), Tubulin (B-5-1-2, Sigma Aldrich, 1:10,000) and subsequently with HRP-tagged secondary antibodies (Jackson Immunoresearch, 1:10,000). Membranes were revealed with the ECL2 substrate according to the manufacturer's instructions (Pierce). Chemiluminescence signals were acquired using the ChemiDoc imaging system (Bio-Rad, Image Lab Touch Software 2.4.0.03) and analysed with ImageLab (Bio-Rad, Image Lab 6.0.1 build 34). Uncropped gels are provided as a Source data file.

## Reporting summary

Further information on research design is available in the Nature Portfolio Reporting Summary linked to this article.

## Data availability

The coordinates and EM maps generated in this study have been deposited in the Protein Data Bank and the Electron Microscopy Data Bank (summarised in Table 1): Influenza polymerase A/H7N9-4M (ENDO(R) | Core 1) PDB 8RMP, EMD-19366. Influenza polymerase A/H7N9-4M (ENDO(R) | Core 2) PDB ID 8RMQ, EMD-19367. Influenza polymerase A/H7N9-4M replication complex, an asymmetric polymerase dimer bound to human ANP32A PDB ID 8RMR, EMD-19368. Influenza polymerase A/H7N9-4M replicase minus 627(R) (from 'Influenza polymerase A/H7N9-4M replication complex' | Local refinement) PDB ID 8RMS, EMD-19369. Influenza polymerase A/H7N9-4M encapsidase plus 627(R)/human ANP32A (from 'Influenza polymerase A/H7N9-4M replication complex' | Local refinement) PDB ID 8RN0, EMD-19382. Influenza B polymerase, monomeric encapsidase with 5' cRNA hook bound PDB ID 8RN1, EMD-19383. Monomeric apo-influenza B polymerase, encapsidase conformation PDB ID 8RN2, EMD-19384. Pseudo-symmetrical influenza B polymerase apo-dimer, encapsidase moiety (from 'Influenza B polymerase pseudo-symmetrical dimer' | Local refinement) PDB ID 8RN3, EMD-19385. Pseudo-symmetrical influenza B polymerase apo-dimer, ENDO(T) moiety (from 'Influenza B polymerase pseudo-symmetrical dimer' | Local refinement) PDB ID 8RN4, EMD-19386. Pseudo-symmetrical influenza B polymerase apo-dimer, ENDO(R) moiety (from 'Influenza B polymerase pseudo-symmetrical dimer' | Local refinement) PDB ID 8RN5, EMD-19387. Pseudo-symmetrical influenza B polymerase apo-dimer, ENDO(E) moiety (from 'Influenza B polymerase pseudo-symmetrical dimer' | Local refinement) PDB ID 8RN6, EMD-19388. Pseudo-symmetrical influenza B polymerase apo-dimer, core-only moiety (from 'Influenza B polymerase pseudo-symmetrical dimer' | Local refinement) PDB ID 8RN7, EMD-19389. Influenza B polymerase pseudo-symmetrical apo-dimer (FluPol(E)|FluPol(S)) PDB ID 8RN8, EMD-19390. Influenza B polymerase, replicase (from 'Influenza B polymerase apo-trimer' | Local refinement) PDB ID 8RN9, EMD-19391. Influenza B polymerase, encapsidase plus 627(R)/human ANP32A (from 'Influenza B polymerase apo-trimer' | Local refinement) PDB ID 8RNB, EMD-19393. Influenza B polymerase, replication complex, an asymmetric polymerase dimer bound to human ANP32A (from 'Influenza B polymerase apo-trimer' | Local refinement) PDB ID 8RNC, EMD-19394. Influenza B polymerase apo-trimer PDB ID 8RNA, EMD-19392. The cryo-EM raw data (10.15151/ESRF-ES-1299132129 and 10.15151/ESRF-ES-1324918289) are available under restricted access for a period of 3 years (ESRF embargo). Access can be obtained before the end of this embargo by request to S.C. or after the end of this embargo through the ESRF's DOI data portal (https://data.esrf.fr/doi/). Source data are provided with this paper.

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

## Acknowledgements

We thank Aldo R. Camacho-Zarco for providing hANP32A truncated constructs proteins. We thank Wojtek Galej, Sarah Schneider and Romain Linares for access to the Glacios at EMBL Grenoble; Aymeric Peuch for support using the joint EMBL-IBS computer cluster and Caroline Mas help with the Mass Photometer measurements. We acknowledge the European Synchrotron Radiation Facility and Grenoble Partnership for Structural Biology for access to the Titan Krios CM01 and Romain Linares and Gregory Effantin for assistance with data collection. We thank Matthias Budt and Thorsten Wolff (Robert Koch Institute, Berlin) for providing the HEK-293T ANP32AB KO cells and Sylvain Paisant (Institut Pasteur) for helping with plasmid mutagenesis and purification. This work was partially funded by the ANR grant FluTranscript (ANR-18-CE18-0028), held jointly by S.C. and N.N. T.K. was funded by the ANR grants FluTranscript and ANR-10-LABX-62-IBEID. This work used the platforms at the Grenoble Instruct-ERIC Center (ISBG; UMS 3518 CNRS CEA-UGA-EMBL) with support from the French Infrastructure for Integrated Structural Biology (FRISBI; ANR-10-INSB-05-02) and GRAL, a project of the University Grenoble Alpes graduate school (Ecoles Universitaires de Recherche) CBH-EUR-GS (ANR-17-EURE-0003) within the Grenoble Partnership for Structural Biology.

## Author contributions

S.C. and B.A. conceived the project. M.P. and B.A. performed cloning. P.D. did initial biochemical studies on FluPol-hANP32A complexes. B.A. performed all other in vitro biochemical, biophysical and cryo-EM analyses. S.C. and B.A. did model building and refinement. Discussions with M.B. led to inclusion of Fig. 3I. T.K., supervised by N.N., performed all cellular assays. S.C. and B.A. prepared the manuscript with input from all authors, especially T.K. and N.N.

## Funding

## Competing interests

The authors declare no competing interests.
