## [Peer Review File · Nature Communications]

Structures of influenza A and B replication complexes give insight into avian to human host adaptation and reveal a role of ANP32 as an electrostatic chaperone for the apo-polymeraseREVIEWER COMMENTS

Reviewer #1 (Remarks to the Author):

Arragain et al compare the replicase-encapsidase structures of the influenza A and B RNA polymerases, and the role of ANP32 in stabilizing the complexes. The authors use size-exclusion chromatography and mass photometry analyses to show that ANP32 binding to the influenza A and B virus RNA polymerase is salt-dependent. In addition, the authors use cryo-EM to reveal the structures of new polymerase-polymerase interactions. The new multimers are, according to the authors themselves "likely too rare to be detected in the biochemical and biophysical experiments". The description of these new, potentially rare or maybe transient interactions forms the core of the manuscript and is an interesting addition to the field.

Overall, the study is complex. It confirms and ties together several previous studies by the Cusack, Fodor, Naffakh, and Barclay labs: that promoter binding by the influenza virus RNA polymerase increases solubility and leads to monomer formation, while binding of ANP32 improves dimerization and RNA synthesis in several ways. However, the manuscript is also relatively dense, and I worry that it may be hard to follow for non-flu-RNA-polymerase-experts in the field.

Specific comments:

1. The justification for using a mutant H7N9 for comparison with the influenza B virus RNA polymerase, and not the wildtype H7N9, is not clear. Moreover, one of the key conclusions of the manuscript focuses on the differences between the influenza types, but a direct mutant-to-mutant or wildtype-to-wildtype comparison is not made. At least this could be recognised as a potential limitation of the study and should be explained better.
2. The authors claim that formation of the asymmetric dimer formation by the H7N9 and influenza B virus RNA polymerase requires ANP32 at 150 mM salt. However, controls showing the behaviour of the RNA polymerase alone at 300 mM and 150 mM salt concentrations are not shown in Fig. 1 or S1. The polymerase may not be sufficiently soluble at lower salt concentrations on its own, which is hinted at, but this could be clarified or shown.
3. The models and conclusions may have impacts on our understanding of influenza virus replication, beyond what is discussed. If the polymerase is not soluble in apo-form and immediately forms/needs to form dimers upon interacting with ANP32, do the authors suggest that stable, monomeric trimers cannot exist in a cell (intracellular ionic strength is <300 mM). And, taking that one step further, do they suggest that if free trimers do not exist, then trimers must be assembled in the nucleus as dimers of trimers, and that ANP32 is the chaperone to help this assembly? These seem testable ideas or at least points for discussion.
4. Figure 7 is difficult. The flow goes from A to C to D to B to E. This is confusing. The mixing of FluPolA and FluPolB does not help the clarity either. The intermediate state C does not seem logical (must the RNP dissociate from the CTD just to undergo a conformational change?) and appears to contradict the model proposed in the thorough study by Krischuns et al 2024. One can imagine C depicting incoming RNPs, but in that case it should be labeled A.
5. Line 249-250; for clarity, it might be better to start the description of the mutagenesis with a new paragraph.
6. The manuscript appears to be formatted for a Cell/Elsevier submission. For instance, the methods are formatted as STAR methods and the references are not numbered.

Reviewer #2 (Remarks to the Author):

Arragain et al. determined the cryo-EM structures of the replication complexes for influenza A and B. Specifically, they described host factor ANP32 and its interactions with FluPol. Binding assays, mass photometry, and vRNP reconstitution assays complemented the cryo-EM data.

The manuscript describes how ANP32 is essential to prevent aggregation of FluPolA and FluPolB in addition to forming dimers. Observing the FluPol in complex with ANP32 in low salt conditions but not in high salt conditions is intriguing. In addition, cryo-EM structures show how ANP32 stabilizes

the replication complex, and Arragain et al. thoroughly describe their interaction. They also used vRNP reconstitution assays with various mutants to support these claims and draw comparisons to other literature.

Overall, the manuscript is comprehensive and thorough. However, there are some issues that can be addressed.

Major Issues:

1. Observing the FluPol in complex with ANP32 in low salt conditions but not in high salt conditions is intriguing. Arragain et al. tested the 500 mM, 300 mM, and 150 mM NaCl. How about even lower salt conditions, such as 100 mM or 50 mM NaCl?

2. This paper reported the cryo-EM structures of FluA and FluB polymerases in complexes with ANP32 in different states. The cryo-EM maps fitting with some important sites of the model should be presented to indicate the quality of the maps, such as the interfaces between polymerase and ANP32, especially the LCAR of ANP32 (including the maps for Figure S5 B-D). In addition, the particles used for some final maps are less than 100,000; the Fourier Completeness and the angular distribution map of particles used for refinement should be included.

3. Shortening certain sentences or breaking them into two sentences will help improve readability, as some sections are a bit unclear. Sometimes, the subject or predicate of a sentence can be hard to identify. For example, lines 255-258, 267-271, and 299-303 could benefit from rephrasing.

Minor Issues:

1. On page 8, lines 172-176 describe cryo-EM data in support of FluPolB trimers. This data can be moved to a later section because the section is about ANP32 as an electrostatic chaperone.

2. Line 178 mentions that FluPol A/H7N9 forms symmetric dimers, but lines 157-158 mention that it does not form symmetric dimers.

3. In the discussion, line 654 mentioned that FluA can be explained by a trimer model of replication. However, before this line, the manuscript only describes FluB in the context of forming trimers. It could be helpful to include the Fan et al. 2019 citation earlier in the introduction to provide more context.

4. There are a few minor spelling or punctuation errors, but they do not affect comprehension.

5. What is the physiological role of ANP32 in host cells? And the difference among ANP32A, ANP32B, and ANP32E.

6. The authors showed the trimer of FluPolB. How about FluA or FluC polymerases? Are any trimer particles identified in FluAPol data?

7. How did the authors identify the Mg ion in the cryo-EM map?

Reviewer #3 (Remarks to the Author):

This manuscript has investigated the replication mechanisms of influenza A and B viruses, and revealed that ANP32 molecule acts as the electrostatic chaperone to stabilize the encapsidase moiety in the apo-polymerase symmetric dimer which are distinct for influenza A and B polymerases. The encapsidase and ANP32 together will form asymmetric complex with the replicase which is similar as the influenza C replication complex that was previously reported. The cryo-EM structure of fluB replication complex showed that ANP32 LCAR would wrap around and stabilize the apo-encapsidase conformation. They also proposed a polymerase trimer model for the influenza replication. Overall, the concept advance is limited, and the authors have showed a similar asymmetric encapsidase-ANP32-replicase model for the influenza A and B viruses. But the authors have given more insight into how the ANP32 stabilizes the apo-polymerase at physiological salt concentrations. Furthermore, they showed a novel symmetric fluB polymerase dimer structure. They also explained how the known mutations in the polymerase help to adapt to from avian to mammalian host, corresponding to interaction with distinct ANP32 molecules. Several suggestions are summarized below.

1. The manuscript is a bit redundant, and the authors can briefly describe the structural details which is similar among the influenza A, B and C polymerases, and focus on the distinct features for influenza A and B replication complexes.
2. As to the polymerase trimer structure, the physiological significance should be carefully examined. Previous studies revealed that the replicase polymerase resides on the one end of vRNP or cRNP, and the symmetric dimer mainly functions at the replication event from cRNP to vRNP. Therefore, the authors should be cautious that this trimer model indeed exists physiologically, or it is artifact.
3. Figure 7 looks complicated and is hardly to be understood. I suggest separating the fluA and fluB replication complex for the model description or make a common replication model for both influenza A and B viruses. Current figure is confused.
4. The authors should realize that A/H7N9-4M polymerase mutant protein was used for the biochemical and structural analysis. It is interesting that how ANP32 binding balance the ratio between the symmetric apo-polymerase dimer and asymmetric replicase-encapsidase dimer complex. I think the situation will be different for different types of influenza viruses, and the author can discuss it.
5. Lines 169-170, the authors claimed that "the stable FluPolB dimer detected by mass photometry only requires the LCAR". This result is in contradiction to the FluPolA which required LRR together with half the LCAR to form dimer. Thus, I guess this dimer was induced by the low salinity. It is better to perform a control to see whether the low salt concentration could promote the formation of FluPolB dimer.
6. Lines 181-182 and lines 167-168, the authors claimed that apo-FluPol required the ANP32 to maintain stable in physiological salt concentrations. However, according to my experience, the FluPol alone could be much stable in physiological salt concentrations when the protein concentration of FluPol is lower than 0.5 mg/ml. The authors should amend their description.
7. Line 302, an extra comma.
8. Lines 473-475, as mentioned above, the authors could not exclude the possibility that the formation of dimeric FluPolB was induced by low salt concentration.

Summary of major changes to the manuscript are:

- (1) As suggested by several referees, we have added additional *in vitro* size-exclusion chromatography and mass photometry experiments concerning the salt dependence (including down to 50 mM salt) of the binding of hANP32A to apo-FluPolA (monomeric FluPolA/H7N9-4M mutant and dimeric FluPolA/H7N9 wild-type), apo-FluPolB, and the corresponding FluPol oligomerisation state. The full results are now presented in revised “Figure 1” (FluPolA/H7N9-4M and FluPolB, salt dependence), “Supplementary Figure 1” (FluPolA/H7N9-4M and FluPolB, dependence on hANP32A domain), and “Supplementary Figure 2” (FluPolA/H7N9-WT, salt dependence). The full response is given in the reply to Referee 1 point 2 and for ease of reference, a figure summarising the results is also provided within the rebuttal letter.
- (2) To emphasise that the FluPolA and FluPolB symmetric dimers are quite different, a new comparative panel “H” of these two dimers has been added to Figure 4.
- (3) As requested by Referee 2 point 2, 3D-FSCs, particle orientation distribution and map quality are now shown in new “Supplementary Note 3” and “Supplementary Note 7”. Cryo-EM density has also been added to several figures (Supplementary Figure 5; new Supplementary Notes 3, 7).
- (4) Since we lack *in vivo* validation, we have relocated our hypothetical trimer model of replication (Figure 7) to the Supplementary Figures to emphasise its speculative nature. It has been modified in the light of the referee’s criticisms and divided into separate “Supplementary Figure 10” and “Supplementary Figure 11” summarizing the model for FluPolA and FluPolB, respectively.
- (5) We have extensively rewritten the manuscript, to shorten sentences, enhance clarity, improve consistency of nomenclature and reduce confusion.

Reviewer #1 (Remarks to the Author)

Arragain et al compare the replicase-encapsidase structures of the influenza A and B RNA polymerases, and the role of ANP32 in stabilizing the complexes. The authors use size-exclusion chromatography and mass photometry analyses to show that ANP32 binding to the influenza A and B virus RNA polymerase is salt-dependent. In addition, the authors use cryo-EM to reveal the structures of new polymerase-polymerase interactions. The new multimers are, according to the authors themselves “likely too rare to be detected in the biochemical and biophysical experiments”. The description of these new, potentially rare or maybe transient interactions forms the core of the manuscript and is an interesting addition to the field.

Overall, the study is complex. It confirms and ties together several previous studies by the Cusack, Fodor, Naffakh, and Barclay labs: that promoter binding by the influenza virus RNA polymerase increases solubility and leads to monomer formation, while binding of ANP32 improves dimerization and RNA synthesis in several ways. However, the manuscript is also relatively dense, and I worry that it may be hard to follow for non-flu-RNA-polymerase-experts in the field.

Specific comments:

1. The justification for using a mutant H7N9 for comparison with the influenza B virus RNA polymerase, and not the wildtype H7N9, is not clear. Moreover, one of the key conclusions of the manuscript focuses on the differences between the influenza types, but a direct mutant-to-mutant or wildtype-to-wildtype comparison is not made. At least this could be recognised as a potential limitation of the study and should be explained better.

The referee is correct that ideally one would have used the wild-type (WT) FluPolA/H7N9 to investigate the FluA replication complex. However, prior to working on the FluB replication complex, we were only aware that apo-FluPolA/H7N9-WT forms constitutive symmetrical dimers, which previous results suggested compete with the formation of the asymmetric replication complex (Chen *et al.*, 2019; Sheppard *et al.*, 2023). We now know that both FluPolA and FluPolB form (different) symmetrical dimers as well as the asymmetric replication complex.

Furthermore, monomeric mutants of FluPolA, unable to form symmetrical dimers, have been shown to be functional for replication in cell-based assays (e.g. the FluPolA/H7N9-4M mutant used here, Krischuns *et al.*, 2024) and are indeed selected for in the presence of only ANP32E (Shepard *et al.*, 2023). Therefore, we chose to use the FluPolA/H7N9-4M in our studies on the FluA replication complex, with the aim of eliminating symmetrical dimers and, hopefully, obtaining more asymmetric dimer particles on cryo-EM grids which are, in any case, rare. This strategy was in fact successful.

Subsequently we found that WT apo-FluPolB also forms a symmetric dimer, notably in the presence of hANP32A. Using mass photometry and on grids we observe mainly these symmetrical FluPolB-type dimers, but also some trimers which exhibit both the symmetrical and the asymmetrical dimer interfaces, the latter corresponding to the FluB replication complex.

It has been proposed that FluPolA could also form such a trimer, with both symmetric and asymmetric interfaces, and that this could be important for realignment during cRNA to vRNA replication (Fan *et al.*, 2019, Carrique *et al.*, 2020). However, this trimer has not yet been observed structurally (nor one for FluPolC) and structural validation of the FluPolA trimer hypothesis and the relevance of the FluPolB trimer are subjects for future research.

In the text, we have improved the clarity of the rationale for using the monomeric FluPolA/H7N9-4M mutant. On the other hand, we do not think that the use of this mutant detracts from the comparison of the FluA, FluB and FluC replication complexes.

2. The authors claim that formation of the asymmetric dimer formation by the H7N9 and influenza B virus RNA polymerase requires ANP32 at 150 mM salt. However, controls showing the behaviour of the RNA polymerase alone at 300 mM and 150 mM salt concentrations are not shown in Fig. 1 or S1. The polymerase may not be sufficiently soluble at lower salt concentrations on its own, which is hinted at, but this could be clarified or shown.

We thank the referee for suggesting these experiments. Our response is also valid for Referee 2 point 1 and Referee 3 points 5 and 6.

We have repeated size-exclusion chromatography (μM FluPol concentration) and mass photometry (nM FluPol concentration) experiments to include more conditions (notably at 100 and 50 mM salt) and the controls requested by Referee 1. New data have been included in Fig 1 and Supplemental Figure 2. A summary figure for FluPolA (A/H7N9-WT and monomeric A/H7N9-4M mutant) and FluPolB is included in this rebuttal for ease of reference. These results reveal a complex protein concentration, salt concentration and hANP32A dependent oligomerisation behaviour of both FluPolA and FluPolB. Overall, they support our conclusion that, at physiological salt (~ 150 mM), hANP32A interacts with apo-FluPol and mainly stabilises (pseudo)-symmetrical dimers, at least *in vitro*. We think that these hANP32A bound dimers are a major form of WT apo-FluPol in the nucleus, prior to any encounter with the RNPs that are already there, whether or not ANP32 plays an essential chaperone role *in vivo*.

With reference to the summary Figure at the end of this rebuttal:

(1) FluPolA/H7N9-WT in the absence of hANP32A. The protein is soluble at 500 mM and 300 mM salt as a mixture of mainly monomers and symmetrical dimers according to mass photometry. The structure of the FluPolA/H7N9 symmetrical dimers was described in Kouba *et al.*, 2023. At 150 mM salt, the protein is not soluble at μM concentration in size-exclusion chromatography (SEC) but at nM concentration, mass photometry indicates a mixture of monomers, dimers and tetramers (the latter being previously described by Chang *et al.*, 2015).

(2) FluPolA/H7N9-WT in the absence of hANP32A but presence of the vRNA promoter. The protein is soluble at 150 mM salt and mainly monomeric with some dimers. Structures of these two species were previously described in Kouba *et al.*, 2023.

(3) FluPolA/H7N9-WT in the presence of hANP32A. The protein is soluble in SEC and mass photometry all the way down to 50 mM salt. It is mainly dimeric, but with a small fraction of monomers, increasing at 50 mM salt. Thus, hANP32A interacts and stabilises the apo-FluPolA/H7N9-WT dimer at physiological salt concentrations (150 mM), with 85% dimers compared to 54% in the absence of hANP32A. It is possible that the dimer peak contains some hANP32A-stabilised asymmetric replication complex.

(4) Monomeric FluPolA/H7N9-4M mutant in the absence of hANP32A. The protein is soluble and monomeric at 500 mM and 300 mM salt. At 150 mM salt, it is not soluble at μM concentration in SEC but at nM concentration, mass photometry again indicates soluble monomers.

(5) Monomeric FluPolA/H7N9-4M mutant in the presence of hANP32A. The protein is soluble down to 50 mM salt and mainly monomeric, with a small fraction of dimers. Thus,

hANP32A helps stabilise the monomeric FluPolA/H7N9-4M mutant at physiological salt concentrations (~150 mM). The dimers are likely to be the hANP32A-stabilised asymmetric replication complex, consistent with the low number of such particles seen on cryo-EM grids.

(6) FluPolB in the absence of hANP32A. The protein is soluble and mainly monomeric at 500 and 300 mM salt. At 150 mM salt, the protein is not soluble at μM concentration in SEC but at nM concentration, mass photometry shows that some dimers are formed (14%).

(7) FluPolB in the presence of hANP32A. The protein is monomeric at 500 mM salt and forms dimers at 300 (22%) and 150 (47%) mM salt. At 150 mM salt, the presence of hANP32A stabilises more dimers (47%) than in the absence of hANP32A (14%). Overall, this shows that hANP32A stabilises the apo-FluPolB dimer at physiological salt concentrations, although the dimer can exist without hANP32A under these conditions at nM protein concentration. At 100 and 50 mM salt there are significantly less dimers, the reason for this being unclear.

3. The models and conclusions may have impacts on our understanding of influenza virus replication, beyond what is discussed. If the polymerase is not soluble in apo-form and immediately forms/needs to form dimers upon interacting with ANP32, do the authors suggest that stable, monomeric trimers cannot exist in a cell (intracellular ionic strength is <300 mM). And, taking that one step further, do they suggest that if free trimers do not exist, then trimers must be assembled in the nucleus as dimers of trimers, and that ANP32 is the chaperone to help this assembly? These seem testable ideas or at least points for discussion.

The implications of our *in vitro* observations that both apo-FluPolA/H7N9-WT and apo-FluPolB form different symmetrical dimers, both stabilised by hANP32A at physiological salt concentrations (especially at μM protein concentration), indeed has implications that can only be speculated upon at this stage.

We think that these hANP32A bound dimers are likely the principle form of apo-FluPol in the nucleus, prior to any encounter with the RNPs that are already there. Likely through co-localisation on Pol II pS5 CTD, apo-FluPol-ANP32 dimers will eventually encounter replication-competent RNPs, with which new interactions become possible via transient disassociation of the symmetrical dimers. We propose that one component of the symmetrical apo-dimer, together with ANP32, becomes the encapsidase moiety of the asymmetric replication complex, the promoter-bound replicase being resident in the RNP. The second half of the symmetrical apo-dimer could interact with the asymmetrical replication complex and become the third component of a transient trimer (e.g. involved in cRNA to vRNA replication early realignment, as proposed for FluPolA). ANP32 therefore initially would act as a chaperone for the apo-FluPol symmetrical dimer in the nucleus, note that other chaperones such as HSP90 have been implicated at earlier stages of FluPol biosynthesis, and then as the glue to form the asymmetric replication complex.

We do not see any role for a dimer of trimers. We also do not see a biological role for apo-trimers although they may exist in the nucleus, as we see them *in vitro*, at least for FluPolB. The only functional trimer, would comprise an apo-monomer interacting transiently via a symmetrical interface with an asymmetric replication complex, comprising encapsidase (formerly also an apo-FluPol), ANP32 and replicase, the latter being part of an existing RNP. We have tried to make this clearer in the discussion, as well as the speculative nature of the generalised trimer model.

4. Figure 7 is difficult. The flow goes from A to C to D to B to E. This is confusing. The mixing of FluPolA and FluPolB does not help the clarity either. The intermediate state C does not seem

logical (must the RNP dissociate from the CTD just to undergo a conformational change?) and appears to contradict the model proposed in the thorough study by Krischuns et al 2024. One can imagine C depicting incoming RNPs, but in that case it should be labelled A.

We apologise that the referees found this Figure confusing, particularly the labelling of panel B, which is not in the 'flow' between A and C. On the other hand we point out that the red features labelled as Pol II CTD in A, were also present in C and partially in D, although were unlabelled. Thus there is no implied contradiction with the study of Krischuns *et al*, 2024 (on the contrary), which proposes that replicase binding to the Pol II pS5 CTD is important for replication. It is true that these red features were subsequently omitted in E to H, because it is not known whether this Pol II pS5 CTD binding persists throughout replication and because it is not the focus of this figure.

To take account of these criticisms (and those of Referee 3), we have redrawn Figure 7, separating our hypothetical trimer model of replication for FluPolA and FluPolB, and moved it to Supplementary Figures 10 and 11 respectively, to emphasise its speculative nature.

5. Line 249-250; for clarity, it might be better to start the description of the mutagenesis with a new paragraph.

We have altered the paragraph structure of this section to take account of this suggestion.

6. The manuscript appears to be formatted for a Cell/Elsevier submission. For instance, the methods are formatted as STAR methods and the references are not numbered.

This has been corrected.

Reviewer #2 (Remarks to the Author)

Arragain et al. determined the cryo-EM structures of the replication complexes for influenza A and B. Specifically, they described host factor ANP32 and its interactions with FluPol. Binding assays, mass photometry, and vRNP reconstitution assays complemented the cryo-EM data.

The manuscript describes how ANP32 is essential to prevent aggregation of FluPolA and FluPolB in addition to forming dimers. Observing the FluPol in complex with ANP32 in low salt conditions but not in high salt conditions is intriguing. In addition, cryo-EM structures show how ANP32 stabilizes the replication complex, and Arragain et al. thoroughly describe their interaction. They also used vRNP reconstitution assays with various mutants to support these claims and draw comparisons to other literature.

Overall, the manuscript is comprehensive and thorough. However, there are some issues that can be addressed.

Major Issues:

1. Observing the FluPol in complex with ANP32 in low salt conditions but not in high salt conditions is intriguing. Arragain et al. tested the 500 mM, 300 mM, and 150 mM NaCl. How about even lower salt conditions, such as 100 mM or 50 mM NaCl?

We thank the referee for suggesting these experiments. See answer to Referee 1 point 2, where additional data is presented on the salt-dependent solubility of different FluPol constructs.

2. This paper reported the cryo-EM structures of FluA and FluB polymerases in complexes with ANP32 in different states. The cryo-EM maps fitting with some important sites of the model should be presented to indicate the quality of the maps, such as the interfaces between polymerase and ANP32, especially the LCAR of ANP32 (including the maps for Figure S5 B-D). In addition, the particles used for some final maps are less than 100,000; the Fourier Completeness and the angular distribution map of particles used for refinement should be included.

3D-FSCs, particle orientation distribution and cryo-EM density fitting our model has been included into new “Supplementary Note 3”, for the FluA replication complex and related local refinement structures around the FluPolA replicase and hANP32A-encapsidase moieties, and Supplementary Note 9 for the FluPolB trimer, and related local refinement structures around the FluB replication complex, FluPolB replicase and hANP32A-encapsidase.

3. Shortening certain sentences or breaking them into two sentences will help improve readability, as some sections are a bit unclear. Sometimes, the subject or predicate of a sentence can be hard to identify. For example, lines 255-258, 267-271, and 299-303 could benefit from rephrasing.

These sentences and many others have been rephrased to increase clarity and consistency of nomenclature.

Minor Issues:

1. On page 8, lines 172-176 describe cryo-EM data in support of FluPolB trimers. This data

can be moved to a later section because the section is about ANP32 as an electrostatic chaperone.

We have deleted this reference to the observation of FluPolB trimers in cryo-EM.

2. Line 178 mentions that FluPol A/H7N9 forms symmetric dimers, but lines 157-158 mention that it does not form symmetric dimers.

There is no contradiction here and we apologise that this was unclear. In line 178, we are describing *wild-type* FluPolA/H7N9 (which is well known to form symmetrical dimers) and in lines 157-158, we are describing the *monomeric* FluPolA/H7N9-4M mutant, which was specifically designed to eliminate symmetrical dimers.

3. In the discussion, line 654 mentioned that FluA can be explained by a trimer model of replication. However, before this line, the manuscript only describes FluB in the context of forming trimers. It could be helpful to include the Fan et al. 2019 citation earlier in the introduction to provide more context.

The putative FluPolA trimer (and the Fan *et al.*, 2019 citation) has now been mentioned in lines 135-138, where the proposed role of the FluPolA symmetrical dimer is first introduced.

4. There are a few minor spelling or punctuation errors, but they do not affect comprehension.

We have corrected all those we have seen.

5. What is the physiological role of ANP32 in host cells? And the difference among ANP32A, ANP32B, and ANP32E.

Multiple roles have been assigned to these proteins in chromatin remodelling, apoptosis and nervous system development, most concretely as histone chaperones. A reference to a recent review has been added. Yu, M., Qu, Y., Zhang, H. et al. Roles of ANP32 proteins in cell biology and viral replication. *Animal Diseases* 2, 22 (2022). <https://doi.org/10.1186/s44149-022-00055-7>.

6. The authors showed the trimer of FluPolB. How about FluA or FluC polymerases? Are any trimer particles identified in FluPolA data?

FluPol trimers have only been structurally visualised for FluPolB so far (this work). It has been proposed that FluPolA could also form a trimer, with both symmetric and asymmetric interfaces, and that this could be important for realignment during cRNA to vRNA replication (Fan *et al.*, 2019; Carrique *et al.*, 2020). However, this trimer has not yet been observed structurally, nor any for FluPolC. Structural validation of the FluPolA trimer hypothesis and the relevance of the FluPolB trimer are subjects for future research.

In addition, we did not observe trimers in our mass photometry measurements using FluPolA/H7N9-WT, although it does not necessarily mean it does not exist, as seen for FluPolB.

7. How did the authors identify the Mg ion in the cryo-EM map?

Based on high resolution X-ray and cryo-EM maps, magnesium ions are well-known to exist in FluPol RNA synthesis and PA-ENDO active sites. In the current maps, magnesium ions have been added only where convincing cryo-EM density exists corresponding to the sites observed in higher resolution maps.

Reviewer #3 (Remarks to the Author):

This manuscript has investigated the replication mechanisms of influenza A and B viruses, and revealed that ANP32 molecule acts as the electrostatic chaperone to stabilize the encapsidase moiety in the apo-polymerase symmetric dimer which are distinct for influenza A and B polymerases. The encapsidase and ANP32 together will form asymmetric complex with the replicase which is similar as the influenza C replication complex that was previously reported. The cryo-EM structure of FluB replication complex showed that ANP32 LCAR would wrap around and stabilize the apo-encapsidase conformation. They also proposed a polymerase trimer model for the influenza replication. Overall, the concept advance is limited, and the authors have showed a similar asymmetric encapsidase-ANP32-replicase model for the influenza A and B viruses. But the authors have given more insight into how the ANP32 stabilizes the apo-polymerase at physiological salt concentrations. Furthermore, they showed a novel symmetric FluB polymerase dimer structure. They also explained how the known mutations in the polymerase help to adapt to from avian to mammalian host, corresponding to interaction with distinct ANP32 molecules. Several suggestions are summarized below.

1. The manuscript is a bit redundant, and the authors can briefly describe the structural details which is similar among the influenza A, B and C polymerases, and focus on the distinct features for influenza A and B replication complexes.

We agree with the referee that it is better to be concise but at the same time, details are important as well as validation of presented data with respect to previously published literature or new experimental data, for both FluPolA and FluPolB.

Bearing this in mind, we did try to focus mainly on differences between the FluA and FluB replication complexes. These difference include the visualisation within only the FluB complex of the encapsidase PB1-C/PB2-N helical bundle, the new apo-FluPolB symmetrical dimer, the FluPolB trimer, the path of the hANP32A LCAR, not seen in the FluA and FluC replication complexes.

In addition, the most relevant difference with the FluC replication complex is the juxtaposition of the PB2 627-NLS domains, as shown in Figure 3.

2. As to the polymerase trimer structure, the physiological significance should be carefully examined. Previous studies revealed that the replicase polymerase resides on the one end of vRNP or cRNP, and the symmetric dimer mainly functions at the replication event from cRNP to vRNP. Therefore, the authors should be cautious that this trimer model indeed exists physiologically, or it is artefact.

We agree that any discussion of the relevance of the structurally observed FluPolB trimer is speculative at this stage, since we cannot provide evidence for its existence physiologically, although we do provide some evidence that the FluPolB symmetric dimer exists in cells (Figure 4G).

However, we stress that the structurally observed FluPolA and FluPolB symmetrical dimers are completely different (new Figure 4H). As far as we know, the FluPolA-type symmetric dimer has never been observed for FluPolB and *vice versa*. Therefore, it cannot be assumed that FluPolA and FluPolB behave the same with respect to their symmetric dimers, as seen with their significantly different binding-mode of the Pol II pS5 CTD. As far as we know, there is no evidence that the mechanism whereby realignment of the nascent vRNA during initiation of cRNA to vRNA may be facilitated by transient trimer formation, as proposed for FluPolA (Fan *et al.*, 2019, Carrique *et al.*, 2020), is also valid for FluPolB or indeed FluPolC, where the

FluPolA-type symmetric dimer has not been observed either. Furthermore, recent results have shown that monomeric mutants of FluPolA can perform replication perfectly well in cells (Shepard *et al.*, 2023, Krischuns *et al.*, 2024), shedding some doubt on the essentiality of this mechanism.

In response to this comment, in the revision, we have considerably played down the generalised trimer model that we proposed in the discussion and now moved a revised Figure 7 into Supplementary Figures 10 (FluPolA) and 11 (FluPolB) (see also point 3, below).

3. Figure 7 looks complicated and is hardly to be understood. I suggest separating the FluA and FluB replication complex for the model description or make a common replication model for both influenza A and B viruses. Current figure is confused.

We take the point that this figure is confusing, as also expressed by Referee 1, and we present revised figures where we have separated the FluA and FluB replication cycles and moved them to Supplementary Figures 10 and 11, to emphasise that this model is speculative.

On the other hand, as discussed in the response to point 2, we make no assumptions that FluPolA and FluPolB should have identical mechanisms when it comes to the putative role of a third polymerase.

4. The authors should realize that A/H7N9-4M polymerase mutant protein was used for the biochemical and structural analysis. It is interesting that how ANP32 binding balance the ratio between the symmetric apo-polymerase dimer and asymmetric replicase-encapsidase dimer complex. I think the situation will be different for different types of influenza viruses, and the author can discuss it.

We very much agree with the referee on this point and have been at great pains to emphasise that FluA and FluB (not to speak of FluC) may be different in this respect, given that they have completely different symmetrical dimers, as so far characterised.

5. Lines 169-170, the authors claimed that “the stable FluPolB dimer detected by mass photometry only requires the LCAR”. This result is in contradiction to the FluPolA which required LRR together with half the LCAR to form dimer. Thus, I guess this dimer was induced by the low salinity. It is better to perform a control to see whether the low salt concentration could promote the formation of FluPolB dimer.

We apologise for the confusion here. In lines 169-170, ‘the stable FluPolB dimer’ refers to the FluPolB-specific *symmetrical* dimer, which our results show is principally stabilised by the LCAR. Indeed, we believe that the LCAR is required in particular to stabilise the encapsidase conformation, as supported by our structural observation of the LCAR wrapping around the encapsidase in the FluB replication complex structure.

The “FluPolA dimer” mentioned by the referee, which requires the LRR and half the LCAR, corresponds to the FluPolA *asymmetric* dimer (i.e. the replication complex), since we are here using the monomeric FluPolA/H7N9-4M mutant. The role of the LRR in this complex, to bridge between the encapsidase and the replicase, is very clear from the presented structure of the FluA replication complex. The same holds true for the FluB replication complex. See also our response to Referee 1 point 2 (and the Figure included with these comments), where additional data is presented on the salt-dependent solubility of different FluPols, with or without hANP32A. This shows that the apo-FluPolB symmetrical dimer can exist at 150 mM salt in the absence of hANP32A, but that the latter promotes significantly more dimer

formation, presumably by interacting with and stabilising it. Furthermore, at 100 mM and 50 mM salt, FluPolB dimers appear to dissociate.

6. Lines 181-182 and lines 167-168, the authors claimed that apo-FluPol required the ANP32 to maintain stable in physiological salt concentrations. However, according to my experience, the FluPol alone could be much stable in physiological salt concentrations when the protein concentration of FluPol is lower than 0.5 mg/ml. The authors should amend their description.

We are not sure whether the referee is referring to apo-FluPolA or FluPolB here, or both. In any case, in our hands at μ M FluPol concentration, as used in SEC, both FluPols require hANP32A to be soluble at physiological salt concentration (~150 mM).

For instance, apo-FluPolB alone visibly precipitates (i.e. solution becomes milky) at low salt, but re-dissolves upon adding hANP32A. This probably explains why an apo-FluPolB structure has never been determined up until now. We think this behaviour also illustrates the electrostatic chaperone property of hANP32A, at least *in vitro*.

Whether or not the chaperone effect of ANP32 is important *in vivo* (a topic for future research), it is clear that ANP32 interacts with dimeric apo-FluPol *in vitro* and in cells (see e.g. Figure S3E and J, for split luciferase binding assays in the absence of viral RNA). This probably occurs in the nucleus prior to the apo-FluPol encountering an RNP, as indicated in Supplementary Figures 10 and 11.

7. Line 302, an extra comma.

Removed.

8. Lines 473-475, as mentioned above, the authors could not exclude the possibility that the formation of dimeric FluPolB was induced by low salt concentration.

See answer to point 6.

FluPoIA/H7N9-WT

(1)

500 mM NaCl

300 mM NaCl

150 mM NaCl

(2)

150 mM NaCl + vRNA promoter

150 mM NaCl + hANP32A

(3)

100 mM NaCl + hANP32A

50 mM NaCl + hANP32A

FluPolA/H7N9-4M

(4)

500 mM NaCl

300 mM NaCl

150 mM NaCl

(5)

150 mM NaCl
+ hANP32A

100 mM NaCl
+ hANP32A

50 mM NaCl
+ hANP32A

FluPolB

(6)

500 mM NaCl

300 mM NaCl

150 mM NaCl

(7)

150 mM NaCl + hANP32A

100 mM NaCl + hANP32A

50 mM NaCl + hANP32A

REVIEWERS' COMMENTS

Reviewer #1 (Remarks to the Author):

The authors have addressed my previous concerns in sufficient detail and I am happy to support publication of the manuscript.

Reviewer #2 (Remarks to the Author):

Thank the authors for addressing the initial comments. Following the revision to the article, I do not have more questions now.

Reviewer #3 (Remarks to the Author):

I am satisfied with the revised manuscript, and the authors have answered the majority of the questions. Only one suggestion, as to the trimer model, the authors have agreed that it is speculative and they cannot provide evidence for its existence physiologically, therefore I suggest the authors do not mention it in the abstract, and can only discuss this possibility in the discussion section.